# Cosmogenic $^{10}$Be in river sediment: where grain size matters and why

Renee van Dongen[1], Dirk Scherler[1,2], Hella Wittmann[1] and Friedhelm von Blanckenburg[1,2]

[1]GFZ German Research Centre for Geosciences, Earth Surface Geochemistry, Telegrafenberg, 14473 Potsdam, Germany.
[2]Freie Universität Berlin, Institute for Geological Sciences, 12249 Berlin, Germany.

Correspondence to: Renee van Dongen (dongen@gfz-potsdam.de)

## Abstract

Concentrations of *in situ*-produced cosmogenic [10]Be in river sediment are widely used to estimate catchment-average denudation rates. Typically, the [10]Be concentrations are measured in the sand fraction of river sediment. However, the grain size of bedload sediment in most bedrock rivers cover a much wider range. Where [10]Be concentrations depend on grain size, denudation rate estimates based on the sand fraction alone could potentially be biased. To date, knowledge about catchment attributes that may induce grain size-dependent [10]Be concentrations is incomplete or has only been investigated in modelling studies. Here we present an empirical study on the occurrence of grain size-dependent [10]Be concentrations and the potential controls of hillslope angle, precipitation, lithology and abrasion. We first conducted a study focusing on the sole effect of precipitation in four granitic catchments located on a climate-gradient in the Chilean Coastal Cordillera. We found that observed grain size dependencies of [10]Be concentrations in the most-arid and most-humid catchments could be explained by the effect of precipitation on both the scouring depth of erosion processes and the depth of the mixed soil layer. Analysis of a global dataset of published [10]Be concentrations in different grain sizes (n=73 catchments), comprising catchments with contrasting hillslope angles, climate, lithology and catchment size revealed a similar pattern. Lower [10]Be concentrations in coarse grains (defined as "negative grain size dependency") emerge frequently in catchments which likely have thin soil and where deep-seated erosion processes (e.g. landslides) excavate grains over a larger depth-interval. These catchments include steep (>25°) and humid catchments (>2000 mm yr[-1]). Furthermore, we found that an additional cause of negative grain size dependencies may emerge in large catchments with weak lithologies and long sediment travel distances (>2300-7000 m, depending on lithology) where abrasion may lead to a grain size distribution that is not representative for the entire catchment. The results of this study can be used to evaluate whether catchment-average denudation rates are likely to be biased in particular catchments.

## 1. Introduction

Catchment-average denudation rates are commonly estimated with *in situ*-produced cosmogenic $^{10}$Be concentrations in river sediment (Bierman and Steig, 1996a; Brown et al., 1995; Granger et al., 1996). $^{10}$Be is a rare isotope that is produced within quartz minerals by high-energy cosmic rays in the upper few meters of the Earth's surface (Gosse and Phillips, 2001). Its concentration records the time minerals were exposed to cosmic radiation, which is inversely proportional to denudation rates over time scales of $10^2$-$10^5$ years (Bierman and Steig, 1996b; Brown et al., 1995; Granger et al., 1996; Lal, 1991). Most studies use a sand fraction (0.1-2 mm) of river bedload sediment to estimate catchment-average denudation rates. However, bedload grain sizes found in bedrock rivers, where this method is frequently applied, are often much coarser (Figure 1). The sand fraction provides a representative catchment-average denudation rate only if it is spatially and temporally representative for all erosion sources within the catchment (Bierman and Steig, 1996b; von Blanckenburg, 2005; Brown et al., 1995; Gonzalez et al., 2017; Granger et al., 1996; Neilson et al., 2017; Willenbring et al., 2013). Evaluating this condition is challenging and requires a detailed understanding of the catchment and its erosion processes. Where $^{10}$Be concentrations differ amongst grain size fractions, using a non-representative grain size fraction could bias catchment-average denudation rates by a factor of 3 or more (Lukens et al., 2016). To date, there is no general consensus of what causes grain size-dependent $^{10}$Be concentrations in a catchment. Some studies inferred that lower $^{10}$Be concentrations in coarse grains are caused by deep-seated erosion processes, such as landslides, which excavate material from greater depth where $^{10}$Be concentrations are lower (e.g. Aguilar et al., 2014; Belmont et al., 2007; Binnie et al., 2007; Brown et al., 1995; Puchol et al., 2014; Sosa Gonzalez et al., 2016a, 2016b; Tofelde et al., 2018; West et al., 2014). In a recent study Tofelde et al. (2018) combined a detailed inventory of hillslope processes in a large, semi-arid Andean catchment, with $^{10}$Be concentrations measured in the sand and gravel fraction of river sediment. They explained lower $^{10}$Be concentrations in the gravel compared to the sand by the scouring depth of erosion processes. However, other studies found that grain size reduction by abrasion during fluvial transport, or spatial variations in the provenance of different grains sizes can additionally account for grain size-dependent $^{10}$Be concentrations (Carretier et al., 2009; Carretier and Regard, 2011; Lukens et al., 2016; Lupker et al., 2017; Matmon et al., 2003). This is particularly true for large and high-relief catchments that cover large elevation ranges, because $^{10}$Be production rates depend on atmospheric depth. Carretier et al. (2015) conducted a comprehensive study on $^{10}$Be concentrations in different grain sizes on a precipitation gradient, sampling large Andean catchments. Despite significant contrasts in precipitation and, presumably, weathering and erosion processes, no systematic grain size dependency of $^{10}$Be concentrations as result of precipitation emerged. A reason may be that as catchment size increases, more complexity in controlling factors is added that may blur potential trends in the data (Carretier et al., 2015; Portenga and Bierman, 2011). Hence, it remains elusive which type of catchments are sensitive to grain size-dependent $^{10}$Be concentrations and biased catchment-average denudation rates (e.g., Carretier et al., 2015), or it has only been addressed in modelling studies (e.g. Lukens et al., 2016).

This paper presents the results of an empirical study in which we investigated the occurrence and cause of grain size-dependent $^{10}$Be concentrations in river sediment. Our study consists of two parts: in the first part, we investigated the sole effect of precipitation, in small, granitic catchments in the Chilean Coastal Cordillera that differ mainly by mean annual precipitation. In the second part, we compiled and investigated a global dataset with previously published grain size-dependent $^{10}$Be concentrations to include more catchment attributes in our

analysis. In the following, we first provide a review of processes that control the grain size distribution and [10]Be concentrations of river sediment, to determine relevant catchment attributes for our analysis.

## 2.    Why [10]Be concentrations in river sediment can depend on grain size

### 2.1        Processes that control grain size

The grain size distribution of river sediment is a function of 1) weathering and 2) erosion processes at the hillslope and 3) fluvial processes that change the grain size distribution. Chemical weathering at the hillslope converts bedrock into sediment of different grain sizes at a rate that is controlled by the properties of the parent rock, the climatic regime, and denudation rates at the surface (Sklar et al., 2017). The parent rock mineralogy sets rock dissolution rates and constrains the minimum size of individual minerals. Bedrock fractures provide water pathways and expose fresh bedrock to weathering (Lebedeva and Brantley, 2017; Oberlander, 1972; Ruxton and Berry, 1957). As chemical weathering rates are set primarily by the flux of water flowing through the regolith and, to some degree, also by temperature (Lasaga et al., 1994; Maher, 2010; White et al., 1999), there is a strong dependency of weathering and sediment production on the climatic regime (Dixon et al., 2016; Riebe et al., 2004). The presence of biota in humid climates can enhance the breakdown of rock fragments because microbes play an important role in chemical weathering processes, and growing roots can fracture bedrock (Drever, 1994; Ehrlich, 1998; Gabet and Mudd, 2010; Roering et al., 2010). The size reduction of bedrock fragments in the regolith depends, besides the chemical weathering rate, also on the time they spend in the regolith layer. This regolith residence time is controlled by the thickness of the regolith layer and the denudation rate, i.e., the rate of sediment removal from the surface (Anderson et al., 2007; Attal et al., 2014; Sklar et al., 2017).

Hillslope sediment is transported towards the river channel by a variety of erosion processes. Diffusive processes are considered to operate in slowly eroding, soil mantled landscapes and move relatively fine grains at or near the surface (Roering et al., 1999). In contrast, deep-seated erosion processes (e.g., landslides) are frequent in steep and rapidly eroding bedrock landscapes (e.g., Burbank et al., 1996; Hovius et al., 1997; Montgomery and Brandon, 2002). Especially when a critical threshold hillslope angle of ~25-30° is exceeded, denudation rates are dominated by the frequency of landslides (Larsen and Montgomery, 2012; Montgomery and Brandon, 2002; Ouimet et al., 2009). Deep-seated erosion processes excavate sediment and bedrock fragments of any size from a greater depth interval (Casagli et al., 2003).

The type of erosion process is indirectly controlled by tectonic uplift rates, precipitation and lithology. In a steady state landscape, denudation rates are set by tectonic uplift, which controls river incision and hillslope steepening (DiBiase et al., 2010; Scherler et al., 2014; Whipple and Tucker, 1999). Extreme precipitation events may oversaturate hillslopes and increase the susceptibility to hillslope failure (Chen and Lee, 2003; Gabet and Dunne, 2002). Some authors argue that deep-seated erosion processes are also important in arid landscapes (Aguilar et al., 2014). Finally, the bedrock strength and fracture abundance affects the susceptibility to hillslope failure (e.g., Clarke and Burbank, 2011; Perras and Diederichs, 2014) and constrains how much energy is needed for the detachment and transport of individual particles. Once the sediment has reached the channel, processes like downstream abrasion, selective transport and mixing of sediment sources control the grain size distribution at the sample location.

If mixing within the channel is incomplete, single tributaries or local inputs of sediment (e.g. landslides) may dominate the grain size distribution (Binnie et al., 2006; Neilson et al., 2017; Niemi et al., 2005; Yanites et al., 2009). Downstream abrasion and selective transport result in a progressively smaller grain size distribution. Abrasion wears off the outer layers of clasts (Kodoma, 1994; Sklar et al., 2006) and depends on the travel distance and velocity as well as the lithology of the clasts (Attal and Lavé, 2009). Selective transport

preferentially deposits coarse grains when the transport capacity of water is low (Ferguson et al., 1996; Hoey and Ferguson, 1994) and thus further changes the grain size distribution.

## 2.2    Processes that control variations in [10]Be concentrations

[10]Be concentrations in quartz grains depend on the [10]Be production rate and a grains' exposure time to cosmic rays (Gosse and Phillips, 2001). Processes that preferentially transport sediment from locations with contrasting

[10]Be production rates (i.e. different soil depths or elevations within the catchment) or exposure times (i.e., locations with different denudation rates), result in a larger variation of [10]Be concentrations in the sediment. A certain sample location provides a spatially and temporally representative sediment sample when the sediment from different sources is sufficiently mixed (Binnie et al., 2006; Neilson et al., 2017; Niemi et al., 2005; Yanites et al., 2009). Because [10]Be production rates decrease exponentially with depth (Gosse and Phillips, 2001),

hillslope sediment that is excavated over a larger depth interval by landslides will obtain a larger variation in [10]Be concentrations than sediment transported by diffusive processes near the surface (Aguilar et al., 2014; Belmont et al., 2007; Binnie et al., 2007; Brown et al., 1995; Puchol et al., 2014; Sosa Gonzalez et al., 2016a, 2016b; Tofelde et al., 2018; West et al., 2014). In soil-mantled landscapes, bioturbation by burrowing animals and tree throw (Gabet et al., 2003) result in well-mixed surface layers with uniform [10]Be concentrations (Brown

et al., 1995; Granger et al., 1996; Schaller et al., 2018) (Figure 2). These mixed soil layers are most likely to develop in humid and slowly eroding catchments, where biota is abundant. Eroded sediment from these layers is expected to have uniform [10]Be concentrations. In rapidly eroding and arid landscapes, however, soils are typically very thin or absent, and the eroded sediment likely yield larger variations in [10]Be concentrations (Figure 2). Furthermore, fluvial processes can affect the grain size distribution at the sample location in a way

that not all parts of a catchment are equally represented in different grain size fractions. For example, sediment provenance of grains from different elevations could play a role in catchments with heterogeneous rock types that produce different clast sizes or contain different quartz abundances (Bierman and Steig, 1996b; Carretier et al., 2015). An unequal representation of elevations in different grain size fractions may also result from hydrodynamic sorting, downstream abrasion and insufficient mixing of tributaries that drain different elevations

(Carretier et al., 2009; Carretier and Regard, 2011; Lukens et al., 2016; Neilson et al., 2017). Combined with elevation-dependent [10]Be production rates (and provided that denudation rates are constant), this could also result in grain size-dependent [10]Be concentrations (Carretier et al., 2015; Lukens et al., 2016; Matmon et al., 2003).

## 2.3    Catchment attributes that potentially control grain size-dependent [10]Be concentrations

Based on the above review of processes that may influence grain size-dependent [10]Be concentrations, we identified the catchment attributes mean basin slope, mean travel distance, mean annual precipitation (MAP), and lithology that we will focus on in our study. Here, we consider mean basin slope as a topographic catchment

attribute that controls denudation rates and the scouring depth of diffusive or deep-seated erosion processes. We selected MAP because of its effect on both weathering rates and the scouring depth of erosion processes and lithology because it affects chemical weathering rates, the grain size of individual minerals and the susceptibility to hillslope failure. Finally, we selected mean travel distance of sediment as a metric for fluvial processes that are transport-dependent (e.g. abrasion and hydrodynamic sorting).

## 3. Study area

For our case study, we selected 4 small catchments (<10 km$^2$) located in the Coastal Cordillera of central Chile (Table 1, Figure 3). The Coastal Cordillera features a pronounced latitudinal climate and vegetation gradient, whereas the tectonic setting is rather uniform. The selected catchments are located in the National Park Pan de Azúcar (AZ) (~26° S), the National Reserve Santa Gracia (SG) (~30° S), the National Park La Campana (LC) (~33° S) and the National Park Nahuelbuta (NA) (~38° S). The catchments share a granodioritic lithology, though some minor variations in mineralogy exist between the sites (Oeser et al., 2018). The three northern catchments experience modern uplift rates of <0.1 mm yr$^{-1}$ (Melnick, 2016). The southern-most catchment is located in the Nahuelbuta Range, where uplift rates increased from 0.03-0.04 mm yr$^{-1}$ to >0.2 mm yr$^{-1}$ at $4 \pm 1.2$ Ma (Glodny et al., 2008; Melnick et al., 2009). Because the sampled catchment is located upstream of a river channel knickpoint, it may not yet be influenced by the increased uplift rates (Crosby and Whipple, 2006). The climatic regime and mean annual precipitation (MAP) range from arid (MAP ~13 mm yr$^{-1}$) in Pan de Azúcar in the north, to semi-arid in Santa Gracia (MAP ~88 mm yr$^{-1}$), Mediterranean in La Campana (MAP ~358 mm yr$^{-1}$), and temperate in Nahuelbuta (MAP ~1213 mm yr$^{-1}$) in the south (Meyer-Christoffer et al., 2015). This latitudinal increase in MAP results in an increase in vegetation density. The Normalized Difference Vegetation Index (NDVI) (Didan, 2015) varies from 0.1 in the northern-most catchment to 0.8 in the southern-most catchment (Figure S1). The increase in MAP is accompanied by an increase of chemical weathering rates measured in soil profiles located within or in proximity of the catchments (Oeser et al., 2018). The chemical depletion fraction (CDF), a measure to quantify chemical weathering (Riebe et al., 2003), increases from ~0.1 in Pan de Azúcar (AZ), to ~0.4-0.5 in Santa Gracia (SG), and ~0.3-0.6 in La Campana (LC). Due to heterogeneities in bedrock samples collected in Nahuelbuta (NA), no reliable CDF could be assigned. $^{10}$Be depth profiles measured in two midslope soil profiles, revealed an increasing thickness of the mixed soil layer, presumably due to bioturbation (Schaller et al., 2018). The depth of the mixed soil layer increases from ~0-17.5 cm in Pan de Azúcar (AZ), to ~25-45 cm in Santa Gracia (SG), ~47.5-85 cm in La Campana and ~70 cm in Nahuelbuta (NA) (Schaller et al., 2018). By selecting small and low relief catchments with similar lithology and a relatively uniform tectonic setting, we aim to explore the relationship between grain size and $^{10}$Be concentrations as controlled by precipitation.

## 4. Methods

### 4.1 Sampling and analytical methods

In each of the four Chilean catchments, we collected approximately 6 kg sand and pebbles from the active channel (Figure 3) and conducted a Wolman pebble count (Wolman, 1954) with 1 m intervals to measure the

grain size distribution at the sample locations (Figure S2). We dried and sieved the samples in the laboratory to separate the grain size fractions 0.5-1 mm, 1-2 mm, 2-4 mm, 4-8 mm, 8-16 mm, 16-32 mm and 32-64 mm. Before further processing, we crushed pebbles larger than 1 mm. To isolate pure quartz, we separated and purified the river sediment using standard physical and chemical separation methods (Kohl and Nishiizumi, 1992). We spiked between 10 to 20 g of pure quartz with 0.15 mg $^9$Be carrier, dissolved the quartz and extracted beryllium following established protocols (e.g. von Blanckenburg et al., 2004). Accelerator mass spectrometry measurements were carried out at the University of Cologne, Germany. Reported $^{10}$Be/$^9$Be ratios have been normalized to the KN01-6-2 and KN01-5-3 standards, with nominal $^{10}$Be/$^9$Be ratios of $5.35 \times 10^{-13}$ and $6.32 \times 10^{-12}$, respectively. We calculated $^{10}$Be concentrations from $^{10}$Be/$^9$Be ratios and a blank correction was performed. We used MATLAB® and the CRONUS functions (Balco et al., 2008) with the time-independent (St) scaling scheme (Lal, 1991; Stone, 2000) and the SLHL production rate of 4.01 at g$^{-1}$ yr$^{-1}$ (Borchers et al., 2016; Phillips et al., 2016) to calculate catchment-average denudation rate estimates from the $^{10}$Be concentrations

### 4.2     Global compilation

We compiled data from previously published studies that measured $^{10}$Be concentrations in different grain size fractions sampled at the same location. Because we are interested in small to medium-sized bedrock catchments, and to reduce the effect of long-term floodplain sediment storage in large basins, we discarded basins with an area of >5000 km$^2$. We also removed studies that only measured sand fractions (<2 mm) as weathering and erosion processes affecting these sand-sized grain size fractions may be similar. Hence, we only selected studies measuring at least one sand fraction (mean grain size <2 mm) and at least one coarser grain size fractions (mean grain size >2 mm) (Aguilar et al., 2014; Belmont et al., 2007; Brown et al., 1995; Carretier et al., 2015; Clapp et al., 2002; Derrieux et al., 2014; Heimsath et al., 2009; Matmon et al., 2003; Palumbo et al., 2011; Puchol et al., 2014; Reinhardt et al., 2007; Stock et al., 2009; Sullivan, 2007; Tofelde et al., 2018). From each selected catchment we compiled the reported grain size classes, the corresponding $^{10}$Be concentrations ($\pm$ analytical errors), and the sample location coordinates (Table S3). For studies that reported a grain size fraction as 'larger than', we assumed that the upper grain size limit corresponds to twice the lower limit (e.g. reported: >2 mm, data compilation: 2-4 mm). We acknowledge that this range might be incorrect, but a fixed grain size range was required for proper data analysis. We transformed the measured grain sizes to phi-based grain size classes, which is the negative logarithmic to the base 2 of the grain size diameter (Krumbein, 1934, 1938). The range of grain sizes we investigated (0.063 to 200 mm) corresponds to phi values of -4 to 7.64. To compare data from different study areas with different $^{10}$Be production rates, we normalized the $^{10}$Be concentrations ($\pm$ analytical uncertainties) by the arithmetic mean concentration of all samples from the same catchment.

To assess the influence of the identified catchment attributes mean basin slope, mean travel distance, mean annual precipitation, and lithology (Section 2.3) on grain-size trends in the global compilation, we used a 90-m resolution SRTM DEM (Jarvis et al., 2008). We obtained upstream areas based on the published sample coordinates and using the flow routing tools of the TopoToolbox v2 (Schwanghart and Scherler, 2014). We calculated the topographic parameters: catchment area, mean basin slope, total relief (maximum elevation - minimum elevation) and the mean travel distance of sediment to the sample location, which is calculated as the arithmetic mean travel distance of all pixels in the catchment to the sample location). The agreement between the published and recalculated topographic parameters is good, and minor deviations likely result from

differences in DEM resolution (Figure S3). We obtained an estimate of mean annual precipitation (MAP) in each catchment using the 0.25°-resolution gridded precipitation data set from the Global Precipitation Climatology Centre (Meyer-Christoffer et al., 2015). To classify catchment lithology we used the Global Lithological Map (GLiM; Hartmann & Moosdorf, 2012) together with the lithology reported in the original publications. We defined four different lithological classes: sedimentary, magmatic, metamorphic and mixed (>3 different rock types in a catchment).

Next, we used Sternberg's Law to estimate the extent of abrasion of bedload sediment during fluvial transport, to define a travel distance threshold after which abrasion becomes significant:

$$D(L) = D_0\, e^{-\alpha L} \quad (1)$$

Using equation 1, we calculated the grain size $D$ at the sample location, which is derived from an initial grain size $D_0$ at the source, that travelled distance $L$ and decreased in size at a rate given by the reduction coefficient $\alpha$ (Kodama, 1994; Kodoma, 1994; Lewin and Brewer, 2002; Sklar et al., 2006; Sklar and Dietrich, 2008). The reduction coefficient depends on both grain velocity and lithology. Rocks with low tensile strength reduce faster in size during transport than rocks with high tensile strength (Attal and Lavé, 2009). We chose the reduction coefficients based on literature values for field settings (sedimentary rocks: $\alpha = 0.0003$ m$^{-1}$, magmatic rocks: $\alpha = 0.0002$ m$^{-1}$, metamorphic rocks: $\alpha = 0.0001$ m$^{-1}$), which are typically higher than experimental studies due to different particle collision dynamics and the lack of weathering in experimental studies (Sklar et al., 2006). We considered the effect of abrasion to be negligible when a grain size at the sample location (D) falls in the same phi-grain size class as at its erosion source ($D_0$). E.g., for abrasion to be significant, a grain size of 2 mm at the erosion source, must be smaller than 1 mm at the sample location to fall in a lower phi-grain size class. This results in abrasion thresholds for sedimentary, magmatic, and metamorphic rocks of 2300 m, 3500 m, and 7000 m, respectively. For catchments underlain by mixed lithologies, the abrasion threshold lies between 2300 m and 7000 m (Attal and Lavé, 2009).

We quantified the relationship between grain size and [10]Be concentrations by calculating a 'grain size dependency' for each sample set (Figure 4). This is the slope of a linear fit through the [10]Be concentrations of different grain size classes. To account for uncertainties in [10]Be concentrations and for grain size ranges, we used a Monte Carlo approach (n=10,000) to randomly select a point between the mean ± analytical error [10]Be concentrations and the analysed grain size range. We thus obtained a mean ± standard deviation grain size dependency for each catchment. A positive grain size dependency indicates higher [10]Be concentrations in coarser grains, and vice versa.

Next, we used the Kolmogorov-Smirnov Test (KS-test, 5% significance interval) to test whether particular mean basin slope, MAP or sediment travel distance classes showed a significantly different distribution of grain size dependencies (Kolmogorov, 1933; Smirnov, 1939). Finally, we calculated linear regression statistics between the grain size dependency values and the catchment attributes mean basin slope, MAP and mean travel distance and applied a multivariate linear regression model including the effect of all 3 catchment attributes. We did this for the entire dataset and for each individual lithology. As part of the multivariate statistics, we calculated the relative importance (RI) of all catchment attributes, using the LMG approach (Lindeman et al., 1980), of the 'Relaimpo' R studio-package (Grömping, 2006). This provides the percentage of contribution of each catchment attribute to the multivariate regression model $R^2$.

## 5.  Results

### 5.1  Chilean Coastal Cordillera

The measured $^{10}Be$ concentrations in the most arid catchment (AZ) range from 2.8 to 4.6 $\times 10^5$ atoms (g quartz)$^{-1}$, resulting in catchment-average denudation rates of 5.8 ± 0.7 to 10.1 ± 1.1 mm kyr$^{-1}$ (Table 2). In the semi-arid catchment (SG), the $^{10}Be$ concentrations range from 3.6 to 5.2 $\times 10^5$ atoms (g quartz)$^{-1}$, which corresponds to catchment-average denudation rates of 7.5 ± 0.8 to 11.0 ± 1.4 mm kyr$^{-1}$ (Table 2). The $^{10}Be$ concentrations in the Mediterranean catchment (LC) are a factor 10 lower compared to the other catchments and range from 0.2 to 0.6 $\times 10^5$ atoms (g quartz)$^{-1}$, which results in catchment-average denudation rates of 103.7 ± 12.4 to 384.1 ± 54.5 mm kyr$^{-1}$ (Table 2). The temperate catchment (NA) yielded $^{10}Be$ concentrations ranging from 1.8 to 2.9 $\times 10^5$ atoms (g quartz)$^{-1}$, resulting in catchment-average denudation rates of 24.0 ± 2.6 to 40.2 ± 4.5 mm kyr$^{-1}$ (Table 2). Only the arid (AZ) and Mediterranean (LC) catchments show a consistent, but noisy trend between $^{10}Be$ concentrations and grain sizes. In the arid catchment (AZ), $^{10}Be$ concentrations are decreasing with increasing grain size. The 2σ-variability of $^{10}Be$ concentrations measured in all grain size fractions deviates ±18% from the mean (Figure 5). In the Mediterranean catchment (LC), the $^{10}Be$ concentrations of all grain size fractions vary up to ±40% from the mean and display a noisy but positive grain size dependency, i.e., increasing $^{10}Be$ concentrations with increasing grain size (Figure 5). In both the semi-arid (SG) and temperate catchments (NA), the 2σ-variability in $^{10}Be$ concentrations is low (±12% and ±14%, respectively) and rather unsystematic (Figure 5). The smallest grain size fractions (0.5-4 mm) in the semi-arid catchment (SG) show a decreasing trend, but this trend increases again for coarser grain size fractions (4-32 mm). In the temperate catchment (NA), $^{10}Be$ concentrations are uniform in the five smallest grain size fractions (0.5-16 mm), but this trend breaks down at the two largest grain size fractions (16-64 mm), which have lower $^{10}Be$ concentrations.

### 5.2  Global compilation

The global compilation includes 73 catchments covering a wide range of different hillslope angles, sediment travel distances, MAP and lithologies (Figure 6). Figure 7 shows the data of all catchments, classified in 4 slope classes and colour-coded by lithology. Each box represents the normalized $^{10}Be$ concentrations ± analytical uncertainties and the grain size range of a single sample. Uncertainties in $^{10}Be$ concentrations tend to be larger for samples from steeper catchments (>10°), which may be related to higher denudation rates and therefore lower $^{10}Be$ concentrations. Generally, uncertainties are larger for low $^{10}Be/^9Be$ ratios. In catchments with mean basin hillslope angles <10°, $^{10}Be$ concentrations are relatively similar across all grain size classes. In steeper hillslope classes, coarse grains reveal lower $^{10}Be$ concentrations compared to fine grains, with the largest deviations in catchments with hillslope angles >25° (Figure 7). We discern no pattern related to lithology from this figure but we emphasize that magmatic catchments are more abundant in shallow sloping catchments, whereas metamorphic catchments are more abundant in steep catchments.

Figure 8 (and Figure S4, for plots separated by lithology) shows the grain size dependencies of individual catchments, resulting from the slope of a linear fit to the $^{10}Be$ concentrations of all grain size classes (see methods section and Figure 4). Overall, we observe more sample sets that display significantly (i.e. error bar does not overlap with 0) negative (56.2%) trends in grain size-dependent $^{10}Be$ concentrations, than positive (32.8%; Figure 8). 11.0% of the sample sets have grain size dependencies that are not significantly different

from zero, and thus reveal no grain size dependency. Furthermore, negative grain size dependencies are typically stronger (i.e. higher absolute differences between grain sizes) than positive grain size dependencies.

The calculated grain size dependencies reveal a significant breakpoint at a mean hillslope angle of ~15° (KS-test, Figure 8a). Catchments with mean hillslope angles <15° reveal a distribution with predominantly weak grain size dependencies. Steep catchments with hillslope angles >15° show a wider distribution with predominantly negative grain size dependencies (62.3% significantly negative). 70.0% of the catchments that exceed the threshold hillslope (>25°) have significantly negative grain size dependencies. Our analysis of sediment travel distance shows that the amount and magnitude of negative grain size dependencies slightly increase at longer sediment travel distances (Figure 8b). However, catchments that exceeded the abrasion threshold (sedimentary: 2300 m, magmatic: 3500 m, metamorphic: 7000 m, mixed: 2300-7000 m) show no significantly different grain size dependency distribution based on the KS-test. Finally, the data suggests a slightly increasing amount and magnitude of negative grain size dependencies with increasing MAP. Humid catchments (MAP >2000 mm yr$^{-1}$) reveal a distribution of predominantly (90%) significantly negative grain size dependencies, which is significantly different (KS-test) from catchments with MAP <2000 mm yr$^{-1}$ (Figure 8c). However only a low number of catchments with MAP >2000 mm yr$^{-1}$ compose the distribution. Catchments underlain by sedimentary and metamorphic rocks show the most significant negative grain size dependencies (66.7% and 65.4%, respectively), followed by catchments underlain by mixed lithologies (50.0%). The number of significantly negative grain size dependencies is lowest for catchments underlain by magmatic lithologies (37.5%). None of the lithologies revealed a significantly different grain size dependency distribution based on the KS-test.

Linear regressions of grain size dependencies as a function of mean basin slope revealed significantly negative trends for all lithologies combined (p = 0.002) and for metamorphic catchments (p= 0.017) but not for the other lithologies alone (Table S4, Figure S4). MAP showed a significantly negative relationship with grain size dependencies for all lithologies combined (p= 0.007), and for catchments underlain by magmatic (p= 0.006) lithology. However, the trend for magmatic catchments mainly results from one negative data point at higher MAP. No significant linear trends emerged between mean travel distance and grain size dependencies for any of the lithologies.

When considering the combined influence of mean basin slope, MAP and mean travel distance with a multivariate linear model, we found that the variance of all lithologies combined is significantly described (p= 0.004) by two out of all 3 factors, but that the explained variance is low (R$^2$= 0.190, Table S5). Most of the variance is related to mean basin slope (relative importance, RI = 9.1%), followed by MAP (RI = 7.6%), whereas mean travel distance revealed no significant contribution (Table S5, Figure 9). Furthermore, multivariate models yielded significant results when considering only magmatic (p= 0.031, R$^2$= 0.552) and metamorphic catchments (p= 0.077, R$^2$= 0.276). In magmatic catchments, a large proportion of the variability is ascribed to MAP (RI = 51.4%), which results from the above mentioned negative data point at higher MAP. Finally, most of the variability in magmatic catchments was significantly described by mean basin slope (RI = 22.0%). Multivariate statistics yielded insignificant results for mixed and sedimentary lithologies, possibly due to too few catchments to disclose unambiguous trends.

## 6. Discussion

### 6.1 Grain size-dependent $^{10}$Be concentrations in the Chilean Coastal Cordillera

The sampled catchments on the climatic gradient in the Chilean Coastal Cordillera only show a systematic trend of $^{10}$Be concentrations with grain size in the arid (AZ) and Mediterranean catchments (LC). In both catchments, the $^{10}$Be concentrations of river sediment correspond to concentrations measured in the subsurface of the soil profiles (Figure 10; Schaller et al., 2018). Because the difference between $^{10}$Be production rates of the catchment on average and at the soil profiles is small (<10%), we can compare measured $^{10}$Be concentrations

directly. In the arid catchment (AZ), both the negative grain size dependencies and the fact that $^{10}$Be concentrations correspond to concentrations at ~1 m depth in the soil profiles suggest that erosion processes (e.g. rock falls, landslides, gully head retreat), which excavate sediment from intermediate to greater depth during rare precipitation events or earthquakes (e.g. Mather et al., 2014; Pinto et al., 2008), may occur in this catchment. All of the measured $^{10}$Be concentrations in river sediment in the Mediterranean catchment (LC) are

considerably lower than the concentrations measured at the surface of soil profiles in close proximity of catchment (Figure 10; Schaller et al., 2018). This suggests that the catchment experiences faster erosion processes compared to the location of the soil pit, which is confirmed by debris flow scars observed at high elevation in the catchment (Figure S5). Deep-seated erosion processes and insufficient mixing in a small-sized catchment may make a sample non-representative for the entire catchment (Niemi et al., 2005; Yanites et al.,

2009). However, the noisy, but overall positive, grain size dependency in the Mediterranean catchment (LC) contradicts with this hypothesis (Figure S5), as debris flows would presumably excavate coarse grains from greater depth. Higher $^{10}$Be production rates at the elevation where debris flows originate, and the condition that coarse grains only origin from that area cannot account for the positive grain size dependency alone (Figure S5). Without being able to clarify this issue, the lower $^{10}$Be concentrations of river sediment, combined with the

observed greater scatter in the positive grain size dependency may hint at selective transport and longer residence times of coarse grains at higher elevations.

    The $^{10}$Be concentrations in river sediment from the semi-arid (SG) and temperate (NA) catchments show little variations and are similar to concentrations measured near the surface in soil pits (Figure 10; Schaller et al., 2018). Within the temperate catchment (NA), the uniform $^{10}$Be concentrations in grains <16 mm, suggests that

these originate from the ~70 cm thick mixed soil layer, whereas the lower $^{10}$Be concentrations in grains >16 mm suggests these may be derived from below the mixed layer (Figure 10; Schaller et al., 2018). In the semi-arid (SG) catchment, the measured samples from the channel show similar $^{10}$Be concentrations compared to those measured in the mixed soil layer of the north-facing hillslope and higher $^{10}$Be concentrations compared to the mixed layer of the south-facing hillslope (Figure 10; Schaller et al., 2018). This suggests that grains are unlikely

to be derived from greater depth, where $^{10}$Be concentrations are lower.

    We propose that the existing or missing trends in the arid (AZ), semi-arid (SG) and temperate (NA) catchments are mainly related to differences in precipitation and the excavation depth of the erosion processes. These catchments show minor variations in mean basin slope, hence we do not expect big differences in erosion processes due to changes in slope alone. Furthermore, the limited relief of these catchments excludes differences

in $^{10}$Be production rates and local sediment sources to influence observed differences in $^{10}$Be concentrations. However, steeper hillslope angles and higher total relief may have overruled the effect of precipitation in the La Campana catchment. We do not expect a control related to the different catchment sizes in any of the

catchments, because granitic rock have a low abrasion breakdown rate (Attal and Lavé, 2009) and the mean travel distances were small (<1 km).

In summary, we think our new samples from the Chilean Coastal Cordillera suggest an influence of MAP on grain size-dependent [10]Be concentrations only in the most-arid and most-humid catchments by its effect on the thickness of the mixed soil layer and the scouring depth of erosion processes that transport larger grains from below the mixed soil layer.

### 6.2    Grain size-dependent [10]Be concentrations in the global compilation

#### 6.2.1    Mean basin slope

The effect of mean basin slope on grain size-dependent [10]Be concentrations is apparent as weak grain size dependencies in gently sloping catchments, and predominantly negative grain size trends in steep catchments (Figure 7, Figure 8a). Mean basin slope may control grain size-dependent [10]Be concentrations through its effect on the thickness of soils and the scouring depth of erosion processes. In gently-sloping catchments, denudation

rates are typically low (e.g., Portenga and Bierman, 2011) and well-mixed soil layers with uniform [10]Be concentrations can develop. Diffusive erosion processes transport sediment from near the surface, which results in uniform [10]Be concentrations. In contrast, in steep landscapes, denudation rates are usually high and soils are thin or absent if denudation rates exceed the soil production limit (~170 mm kyr[-1]; Dixon and von Blanckenburg, 2012). Such catchments are typically dominated by deep-seated hillslope processes (Hovius et

al., 1997b). Negative grain size dependencies thus occur because coarse grains are excavated from greater depth, where [10]Be concentrations are lower. The highest percentage of negative grain size dependencies are found in catchments steeper than 25°. In these catchments, many hillslopes have likely reached the threshold hillslope angle of ~25-30°, at which denudation rates are dominated by the frequency of landslides (Larsen and Montgomery, 2012; Montgomery and Brandon, 2002; Ouimet et al., 2009). Linear regression models revealed a

stronger control of mean basin slope on grain size-dependent [10]Be concentrations than MAP and mean travel distance, however the $R^2$-values of the regression models were low (Table S5, Figure 9). This conforms with previous studies that also found negative grain size dependencies which emerged from a transition of transport-limited to detachment-limited erosion processes and, therefore, deep-seated erosion processes (Binnie et al., 2007; Brown et al., 1995; Lukens et al., 2016; Reinhardt et al., 2007; Sosa Gonzalez et al., 2016a, 2016b;

Tofelde et al., 2018). It is notable that the most-negative grain size dependencies occur in catchments underlain by sedimentary rocks (Table S4 and Figure S4). This may be due to lower rock mass strength of sedimentary rocks, which partly stems from the presence of bedding planes, making them more susceptible to hillslope failure (e.g., Clarke and Burbank, 2011; Perras and Diederichs, 2014).

#### 6.2.2    Sediment travel distance

Our results revealed a weak negative control of sediment travel distance on grain size dependencies, however no significant relationships were found. The negative control is strongest for sedimentary catchments in which negative grain size dependencies appear to be more frequent in catchments with long sediment travel distances (Figure 8b). For sedimentary catchments the most negative grain size dependencies appear when travel distances exceeded the abrasion threshold. Possibly the lower rock strength of sedimentary rocks promotes the breakdown

into smaller particles and increases the grain's sensitivity to abrasion (Attal and Lavé, 2009; Sklar and Dietrich,

2001). Due to abrasion, distant erosion sources may be overrepresented in finer grain size fractions, and underrepresented in coarser ones (Lukens et al., 2016). As travel distance scales with elevation (Figure S6) and, therefore, $^{10}$Be production rates, sediment from high elevations may have inherently higher nuclide concentrations (Lal, 1991). In contrast, coarse grains, which experienced less abrasion may origin from lower

elevations, with lower $^{10}$Be production rates. This elevation-dependence of certain grain size fractions may induce a negative grain size-dependency. Secondly, if abrasion were to reduce river sediment of decimetre- or meter-scale to sand size, the centre of such clasts would have lower concentrations (Carretier and Regard, 2011; Lupker et al., 2017). However, the associated travel distance has to be considerably longer, and the initial clast must be large. For example, abrasion of an initial 25-cm sized granitic cobble over a distance of ~8 km would

result in a size reduction of 10 cm and expose a centre with a $^{10}$Be concentration that is only 8.5% lower compared to the outer layers (Balco et al., 2008; Sklar et al., 2006). The by-product of abrasion, which typically is of silt or clay size (Sklar et al., 2006), unlikely affects the measured $^{10}$Be concentrations, as it is finer than the grain size classes typically analysed (Lukens et al., 2016). We did not observe a control of sediment travel distance in catchments with mixed lithologies. The provenance of distinct grain sizes from different lithologies

has not resulted in a dominantly positive or negative grain size dependency. Possibly, because the spatial arrangement of different lithologies in a landscape is not necessarily elevation-dependent, or because these lithologies yield minor differences in grain sizes.

### 6.2.3    Mean Annual Precipitation

The global compilation suggested an additional control of MAP on grain size-dependent $^{10}$Be concentrations.

The amount and magnitude of negative grain size trends seems to increase with increasing MAP. The highest percentage of negative grain size dependencies is found in humid catchments (>2000 mm yr$^{-1}$). However this trend is related to a low total number of catchments. Negative grain size dependencies at higher MAP values could be related to higher denudation rates and increasing depth of erosion processes (e.g. precipitation-induced landslides; Chang et al., 2007; Chen et al., 2006; Lin et al., 2008). This differs from our interpretation of the

results from the Chilean Coastal Cordillera, in which we emphasize the control of MAP on the thickness of the mixed soil layer. The discrepancy with the global compilation may result from the additional effect of hillslope angle, which also influences the thickness of the soil mantle and the depth of erosion processes (Heimsath et al., 2009).

### 6.3    Implications

Our results and the above discussion suggest that grain size trends in $^{10}$Be concentrations are best explained by the effects of hillslope angle and MAP on the presence and thickness of mixed soil layers and the scouring depth of erosion processes. In large catchments, an additional effect may emerge by abrasion during transport, which could induce a non-representative grain size distribution. At present, however, it is difficult to quantify the relative roles of hillslope angle, precipitation, travel distance, and lithology, because these parameters tend to be

partly correlated. For example, high and steep topography is often associated with high amounts of orographic precipitation, and long travel distances are associated with high total relief (Figure S6).

In any case, the presumed role that soils and different hillslope erosion processes play for grain size-dependent $^{10}$Be concentrations is likely not linearly related to variables like mean hillslope angle or mean annual

precipitation. Instead, our results are consistent with the presence of thresholds. Landslides likely become important when hillslope angles exceed a critical threshold (Burbank et al., 1996) and once precipitation is high enough to sustain vegetation and soils, diffusive processes may dominate gently-sloping and soil-mantled landscapes. Such a threshold control on the occurrence of grain size-dependent $^{10}$Be concentrations may be the reason why our linear regression statistics, yielded mostly insignificant results or low $R^2$-values (Figure 9 and Table S5). More data may allow better constraining the controls and relative importance of these factors in the future. It additionally highlights the importance of systematic studies on single factors, like our study on the sole effect of MAP in the Chilean Coastal Cordillera.

We evaluated the likelihood of grain size-dependent $^{10}$Be concentrations and a potential bias in previously published $^{10}$Be-derived catchment-average denudation rates, by comparing our findings with a recently published global compilation (Codilean et al., 2018). Out of 2537 different catchments with an area <5000 km², 55.7% have hillslope angles >15°, where our data first shows significant grain size effects, and 23.3% have hillslope angles >25°. When considering sediment travel distances, using the relationship between catchment area and sediment travel distance that emerged from our global compilation ($R^2= 0.99$; Figure S6) about 61.9%, 49.8% and 29.2% of the catchments have exceeded the sediment travel distances of 2300 m, 3500 m and 7000 m, respectively. Finally, 11.5% of the catchments have MAP >2000 mm yr$^{-1}$, based on GPCC-derived MAP at the sample location. Therefore, previously published catchment-average denudation rates may more frequently be biased as a result of steep hillslopes and long sediment travel distance and less frequently by the influence of MAP. When considering a combined effect of all controlling factors in each catchment (slope >25°, sediment travel distance >7000 m and MAP >2000 mm yr$^{-1}$), 49.1% of the catchments are predicted to be devoid of grain size dependencies of $^{10}$Be concentrations and biased catchment-average denudation rates, whereas 50.9% might contain a bias because one or more of the controlling factors has exceeded the threshold values that emerged from our study.

## 7.  Conclusion

In this paper, we used a field study in Chile and a global compilation of previously published data to assess in what type of catchments grain size-dependent $^{10}$Be concentrations may lead to biased estimates in catchment-average denudation rates. Our results suggest that mean basin slope and MAP control grain size-dependent $^{10}$Be concentration through their effect on the presence and thickness of a mixed soil layer and the depth of erosion processes. Hillslope steepness appears to exert the most important influence on grain size-dependent $^{10}$Be concentrations. Our global compilation results show that the influence of MAP is limited to humid catchments (>2000 mm yr$^{-1}$), whereas our case study in Chile suggests an additional control in arid catchments (<100 mm yr$^{-1}$). Furthermore, grain size-dependent $^{10}$Be concentrations may occur in large catchments with long sediment travel distances (>2300 m to >7000 m, depending on lithology), where abrasion may induce non-representative grain size distributions, but this control is less apparent in the current data. We suggest that due to the presence of thresholds, catchment steepness, MAP and sediment travel distance are non-linearly related to grain size-dependent $^{10}$Be concentrations, which complicates efforts to disentangle and quantify their relative roles. The results of our study can be used to evaluate whether catchment-average denudation rates may be biased in particular catchments.

### 8. Dataset availability

All supplementary tables (S1-S5) and figures (S1-S6) are available in the data supplement van Dongen et al., 2019: http://doi.org/10.5880/GFZ.3.3.2019.002. These data are freely available under the Creative Commons Attribution 4.0 International (CC BY 4.0) open access license at GFZ data services. When using the data please cite this paper.

### 9. Sample availability

The metadata of all samples in Table 2 can be accessed via http://igsn.org/[*insert IGSN number here*].

### 10. Author contributions

R. van Dongen carried out fieldwork, laboratory work, and data evaluation. D. Scherler conceived the study and was the main advisor during fieldwork, data evaluation, and manuscript writing. H. Wittmann was responsible for the cosmogenic-$^{10}$Be laboratory training and laboratory supervision. F. von Blanckenburg and H. Wittmann were available for extensive discussion during data evaluation. R. van Dongen prepared the manuscript with contributions and edits from all co-authors.

### 11. Competing interests

The authors declare that they have no conflict of interest.

### 12. Acknowledgements

We acknowledge support from the German Science Foundation (DFG) priority research program SPP-1803 "EarthShape: Earth Surface Shaping by Biota" (grant SCHE 1676/4-1 to D.S.). We are grateful to the Chilean National Park Service (CONAF) for providing access to the sample locations and on-site support of our research. We also thank L. Mao, R. Carrillo and M. Koelewijn for their support during fieldwork, S. Binnie and S. Heinze from Cologne University for conducting AMS measurements, and M. Henehan for his help with statistical analysis. We thank A. Schmidt, S. Tofelde and 2 anonymous reviewers for their constructive reviews.

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

**Table 1: Sample location coordinates and catchment attributes of the research areas located in the Chilean Coastal Cordillera.**

| Catchment | Climate zone | Latitude (°N) | Longitude (°E) | MAP[a] (mm yr$^{-1}$) | Area (km$^2$) | Mean elevation (m) | Total Relief (m) | Mean slope[b] (°) | Mean channel steepness[c] m$^{0.9}$ |
|---|---|---|---|---|---|---|---|---|---|
| Pan de Azúcar (AZ) | Arid | -26.112 | -70.551 | 13 | 0.04 | 339 | 72 | 8.2 | 7.1 |
| Santa Gracia (SG) | Semi-arid | -29.760 | -71.168 | 88 | 0.88 | 773 | 337 | 17.2 | 32.2 |
| La Campana (LC) | Mediterreanean | -32.954 | -71.069 | 358 | 7.41 | 1323 | 1535 | 23.1 | 88.8 |
| Nahuelbuta (NA) | Temperate | -37.808 | -73.014 | 1213 | 5.79 | 1308 | 306 | 8.9 | 20.5 |

[a] Mean annual precipitation (MAP) estimates derived from the GPCC dataset (Meyer-Christoffer et al., 2015).

[b] Total mean basin slope calculated with a 30m DEM.

[c] Normalized channel steepness index.




**Table 2: Cosmogenic nuclide samples from the Chilean Coastal Cordillera. IGSN number, analyzed quartz mass, $^9$Be carries mass, $^{10}$Be/$^9$Be ratio (±1σ), $^{10}$Be concentrations (±2σ analytical error), spallation ($P_{sp}$) and muogenic ($P_{mu}$) production rates and calculated denudation rates (±2σ).**

| Catchment | Grain size (mm) | IGSN | Quartz mass (g) | $^9$Be Carrier mass (mg) | $^{10}$Be/$^9$Be ratio ± 1σ x 10$^{-14}$ | $^{10}$Be concentration ± 2σ (x 10$^5$ atoms g$^{-1}$) | $P_{sp}$ (atoms g$_{qtz}^{-1}$ yr$^{-1}$) | $P_{mu}$ (atoms g$_{qtz}^{-1}$ yr$^{-1}$) | Denudation rate ± 2σ (mm kyr$^{-1}$) |
|---|---|---|---|---|---|---|---|---|---|
| Pan de Azúcar (AZ) | 0.5-1 | GFRD10010 | 9.9 | 0.153 | 43.0 ± 1.5 | 4.48 ± 0.33 | | | 6.04 ± 0.69 |
| | 1-2 | GFRD10011 | 17.9 | 0.153 | 79.9 ± 2.8 | 4.60 ± 0.34 | | | 5.86 ± 0.67 |
| | 2-4 | GFRD10012 | 18.7 | 0.154 | 78.6 ± 5.8 | 4.36 ± 0.32 | | | 6.21 ± 0.72 |
| | 4-8 | GFRD10013 | 18.2 | 0.153 | 65.3 ± 3.6 | 3.69 ± 0.42 | 4.13 | 0.085 | 7.5 ± 1.1 |
| | 8-16 | GFRD10014 | 18.1 | 0.154 | 55.2 ± 2.1 | 3.14 ± 0.24 | | | 8.9 ± 1.0 |
| | 16-32 | GFRD10015 | 15.0 | 0.153 | 40.8 ± 1.5 | 2.80 ± 0.21 | | | 10.2 ± 1.1 |
| | 32-64 | GFRD10016 | 18.7 | 0.153 | 57.6 ± 2.0 | 3.16 ± 0.22 | | | 8.9 ± 1.0 |
| | Mean | - | - | | - | 3.75 ± 0.24 | | | 7.66 ± 0.69 |
| Santa Gracia (SG) | 0.5-1 | GFRD1000Q | 18.7 | 0.154 | 85.4 ± 2.9 | 4.71 ± 0.33 | | | 8.26 ± 0.91 |
| | 1-2 | GFRD1000R | 14.1 | 0.153 | 55.1 ± 2.3 | 4.02 ± 0.34 | | | 9.8 ± 1.2 |
| | 2-4 | GFRD1000S | 13.8 | 0.153 | 49.0 ± 2.1 | 3.62 ± 0.32 | | | 11.0 ± 1.4 |
| | 4-8 | GFRD1000T | 13.8 | 0.153 | 50.3 ± 2.4 | 3.76 ± 0.37 | 6.02 | 0.097 | 10.5 ± 1.4 |
| | 8-16 | GFRD1000U | 20.0 | 0.154 | 82.5 ± 2.7 | 4.25 ± 0.29 | | | 9.3 ± 1.0 |
| | 16-32 | GFRD1000V | 19.3 | 0.154 | 97.0 ± 3.2 | 5.17 ± 0.35 | | | 7.48 ± 0.82 |
| | 32-64 | GFRD1000W | 19.5 | 0.154 | 90.9 ± 3.0 | 4.79 ± 0.33 | | | 8.12 ± 0.89 |
| | Mean | - | - | | - | 4.33 ± 0.26 | | | 9.21 ± 0.84 |
| La Campana (LC) | 0.5-1 | GFRD1000C | 19.4 | 0.154 | 4.98 ± 0.28 | 0.264 ± 0.030 | | | 257 ± 35 |
| | 1-2 | GFRD1000D | 20.0 | 0.154 | 3.44 ± 0.20 | 0.177 ± 0.021 | | | 384 ± 55 |
| | 2-4 | GFRD1000E | 17.0 | 0.154 | 6.05 ± 0.30 | 0.366 ± 0.037 | | | 185 ± 24 |
| | 4-8 | GFRD1000F | 16.9 | 0.154 | 5.70 ± 0.32 | 0.348 ± 0.039 | 9.94 | 0.11 | 194 ± 27 |
| | 8-16 | GFRD1000G | 19.5 | 0.154 | 12.29 ± 0.54 | 0.648 ± 0.059 | | | 104 ± 12 |

| | | | | | | | | | |
|---|---|---|---|---|---|---|---|---|---|
| | 16-32 | GFRD1000H | 20.0 | 0.154 | 9.69 ± 0.44 | 0.498 ± 0.047 | | | 135 ± 17 |
| | 32-64 | GFRD1000J | 16.5 | 0.154 | 9.43 ± 0.43 | 0.588 ± 0.055 | | | 144 ± 14 |
| | Mean | - | - | | - | 0.413 ± 0.033 | | | 200 ± 845 |
| Nahuelbuta (NA) | 0.5-1 | GFRD10002 | 19.8 | 0.154 | 51.4 ± 1.8 | 2.67 ± 0.19 | | | 26.0 ± 2.8 |
| | 1-2 | GFRD10003 | 18.7 | 0.153 | 49.5 ± 2.4 | 2.72 ± 0.27 | | | 25.6 ± 3.3 |
| | 2-4 | GFRD10004 | 18.7 | 0.153 | 51.8 ± 1.9 | 2.84 ± 0.22 | | | 24.5 ± 2.8 |
| | 4-8 | GFRD10005 | 19.2 | 0.154 | 49.7 ± 1.8 | 2.67 ± 0.20 | 10.72 | 0.11 | 26.1 ± 2.9 |
| | 8-16 | GFRD10006 | 20.0 | 0.153 | 56.6 ± 1.9 | 2.90 ± 0.21 | | | 23.9 ± 2.6 |
| | 16-32 | GFRD10007 | 19.6 | 0.154 | 43.5 ± 1.6 | 2.29 ± 0.18 | | | 30.6 ± 3.4 |
| | 32-64 | GFRD10008 | 19.6 | 0.153 | 33.5 ± 1.3 | 1.76 ± 0.14 | | | 40.2 ± 4.5 |
| | Mean | - | - | | - | 2.55 ± 0.16 | | | 27.4 ± 2.4 |

**15. Figure captions**

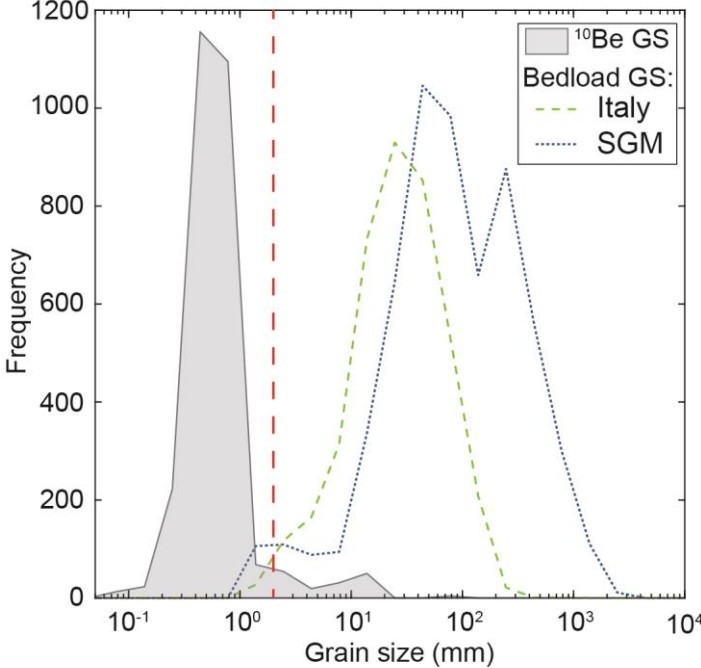


**Figure 1:** Grain size distributions of bedload sediment in rivers and grain sizes used for [10]Be-derived catchment-average denudation rates ([10]Be GS, n=2735) (Codilean et al., 2018). Bedload grain size distributions were measured by pebble counts in bedrock rivers in Southern Italy and Sicily (Italy, n=3900) (Allen et al., 2015; Roda-Boluda et al., 2018) and the San Gabriel Mountains (SGM, n=5930) (DiBiase and Whipple, 2011; Scherler et al., 2016). Wolman
pebble count fractions classified as <2 mm are shown as 1 mm in the figure. Dashed line indicates 2 mm.

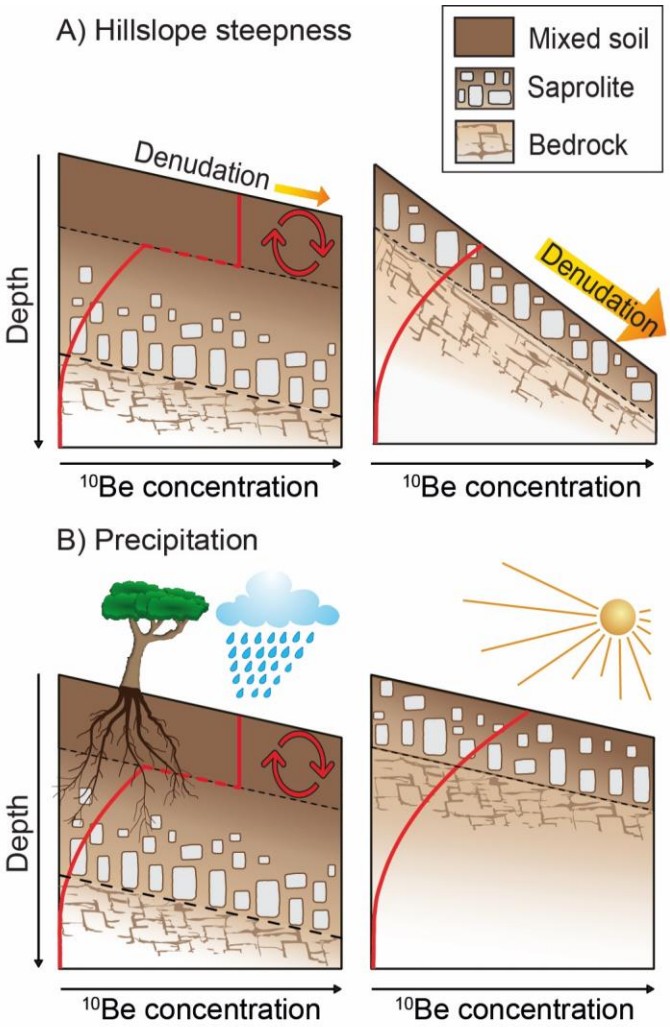

**Figure 2:** The effect of hillslope steepness and precipitation on the depth of the mixed soil layer and [10]Be concentrations as a function of depth. (A) Hillslope steepness and denudation rates control the thickness of the soil-mantle by the removal of material from the top. A thick soil-mantle likely develops in gently sloping and slowly eroding landscapes, whereas high denudation rates in steep landscapes prohibit the development of a thick soil-mantle. (B) Precipitation provides water for chemical weathering. Humid landscapes likely develop a thick soil-mantle, which may be absent in arid landscapes. Bioturbation in landscapes with thick soil-mantles results in a well-mixed soil layer with a uniform [10]Be concentration, which, in isotopic steady state, is equal to the surface concentration. In landscapes where a mixed soil layer is absent, [10]Be concentrations decrease exponentially with depth.

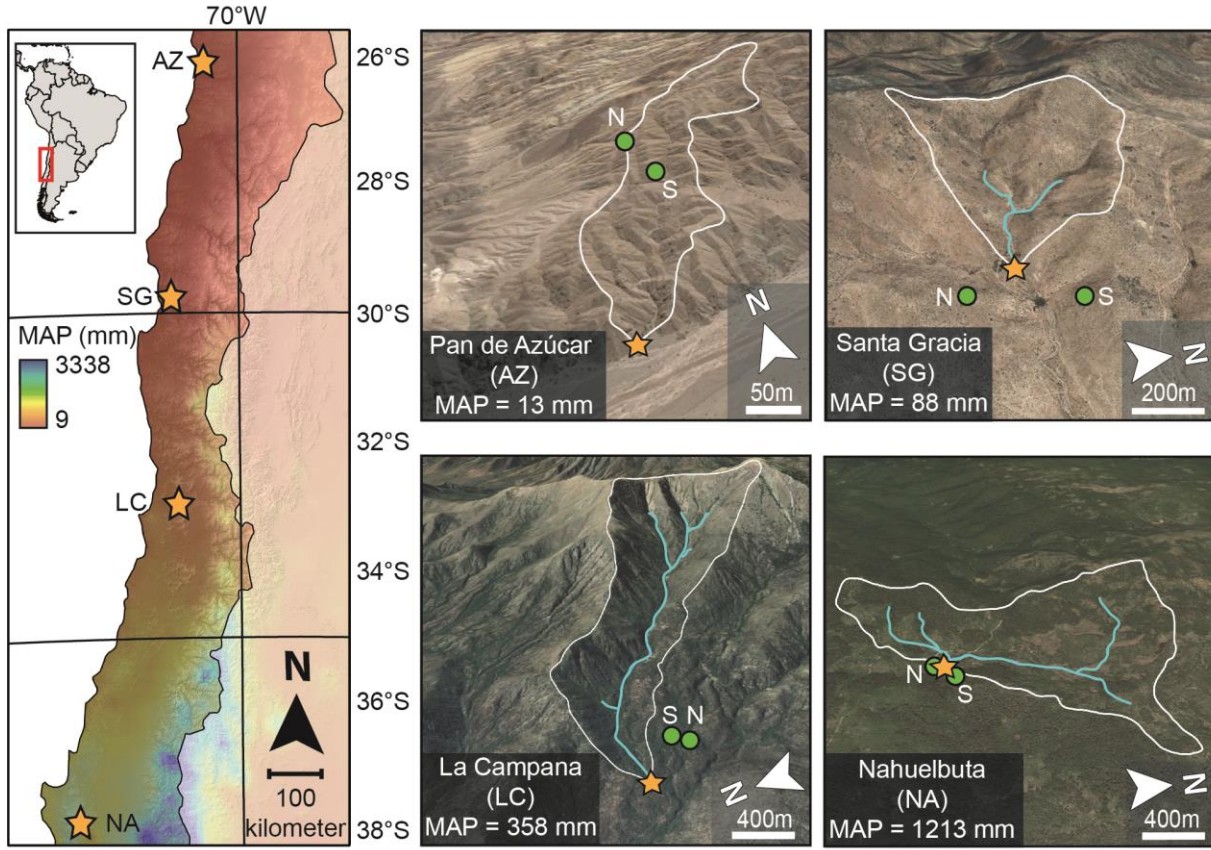


**Figure 3:** Research areas located on a precipitation gradient in the Chilean Coastal Cordillera. On the left, the catchment locations (stars) on a Mean Annual Precipitation (MAP) map from Climate Hazards Group InfraRed Precipitation (CHIRPS), underlain by a SRTM DEM-derived hillshade map. On the right, Google Earth images showing the sample locations (stars), catchment outlines (white), channels with an upstream area of 0.2km$^2$ (blue lines)

and the locations of two [10]Be-depth profiles measured by Schaller et al., 2018 in soil pits on a north-facing (N) and south-facing (S) slope (green dots).

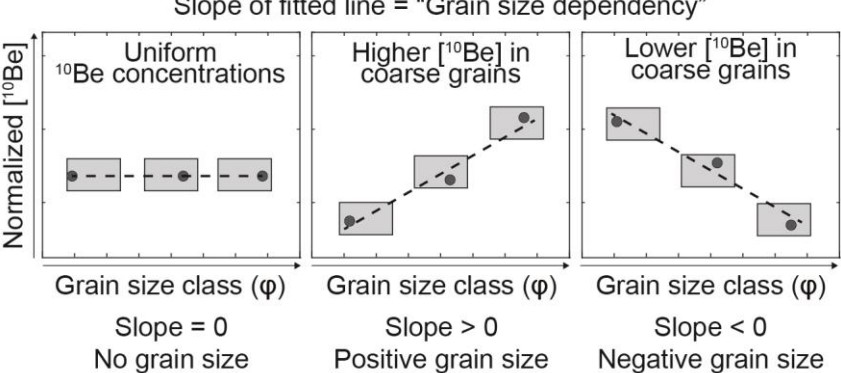

**Figure 4:** Schematic showing the concept of measuring grain size dependency. A random point was selected from within each grain size range and from the corresponding $^{10}$Be concentrations ± analytical error (boxes). The slope of a line fitted to the randomly selected points represents the grain size dependency. We used a Monte Carlo simulation of 1000 runs to account for the width of the grain size range and the analytical errors on $^{10}$Be concentrations. This yields a mean grain size dependency with error bars.


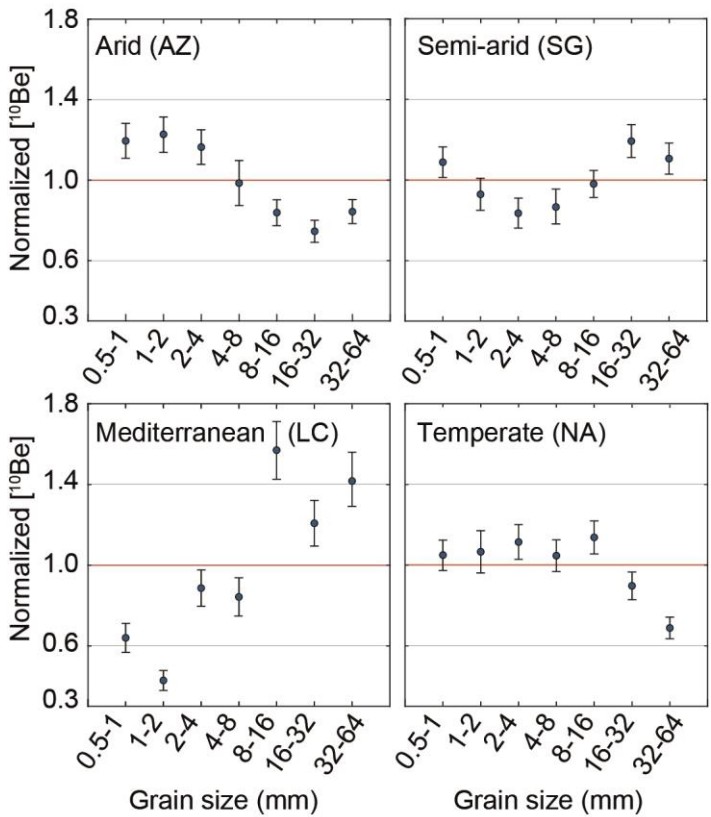

**Figure 5:** Normalized [10]Be concentrations ($\pm 2\sigma$ analytical error) measured in 7 different grain size classes. The [10]Be concentrations are normalized to the arithmetic mean of all grain size fractions within a catchment. The red line indicates the normalized catchment-average [10]Be concentration.

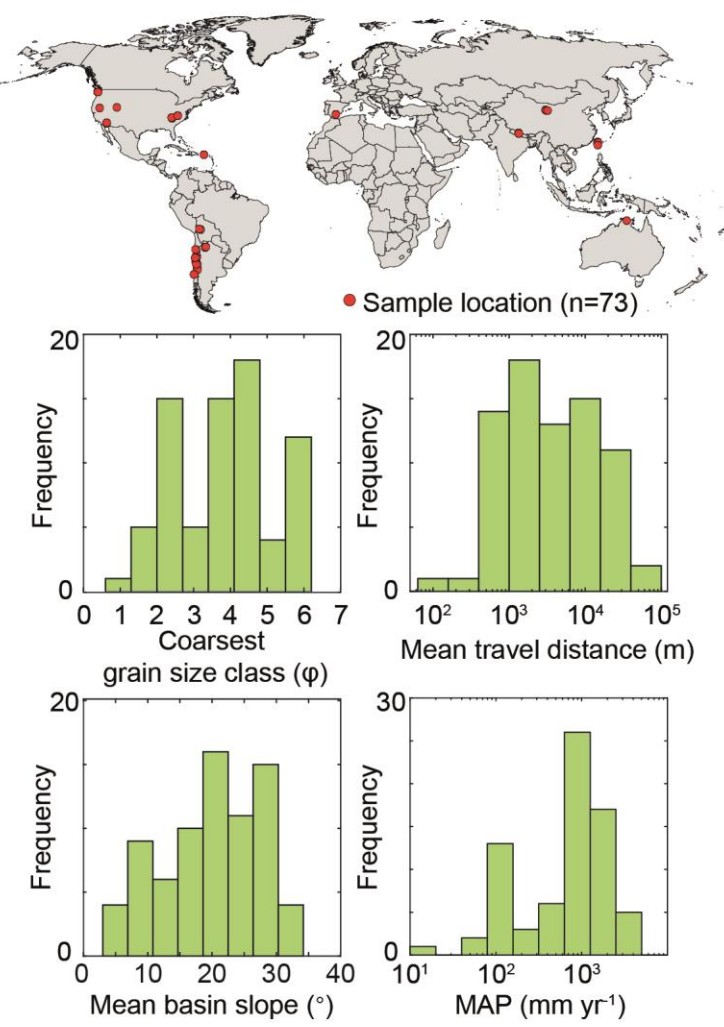

**Figure 6:** Sample locations and catchment attributes of all catchments in the global compilation (n=73). (A) Coarsest phi-grain size classes measured in each study (the smallest grain size was always a sand fraction (<2 mm)). (B) Mean travel distance of sediment, calculated as the arithmetic mean of each grid cell's travel distance towards the catchment outlet. (C) Mean basin slope of each catchment, calculated as the arithmetic mean of the hillslope angles at each grid cell. (D) Mean annual precipitation (MAP) in each catchment, derived from the Global Precipitation Climatology Centre (GPCC) dataset (Meyer-Christoffer et al., 2015).

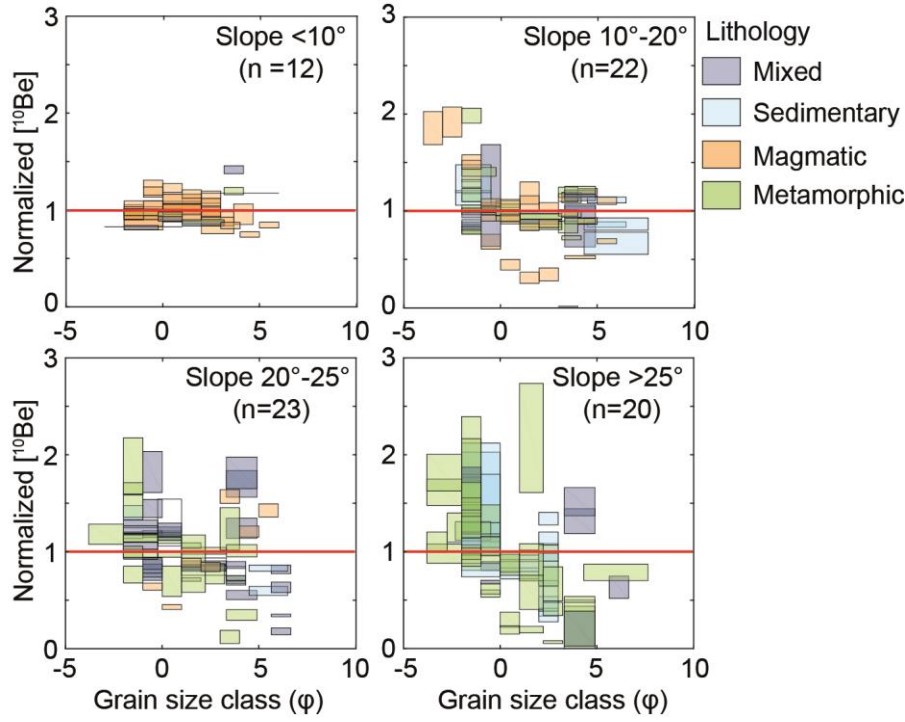

**Figure 7:** All global compilation samples divided into mean basin slope classes and colour-coded by lithology. The boxes indicate normalized mean $^{10}$Be concentrations ± analytical errors and the phi-grain size range. The $^{10}$Be concentrations are normalized by the arithmetic mean $^{10}$Be concentration of all samples from the same catchment.


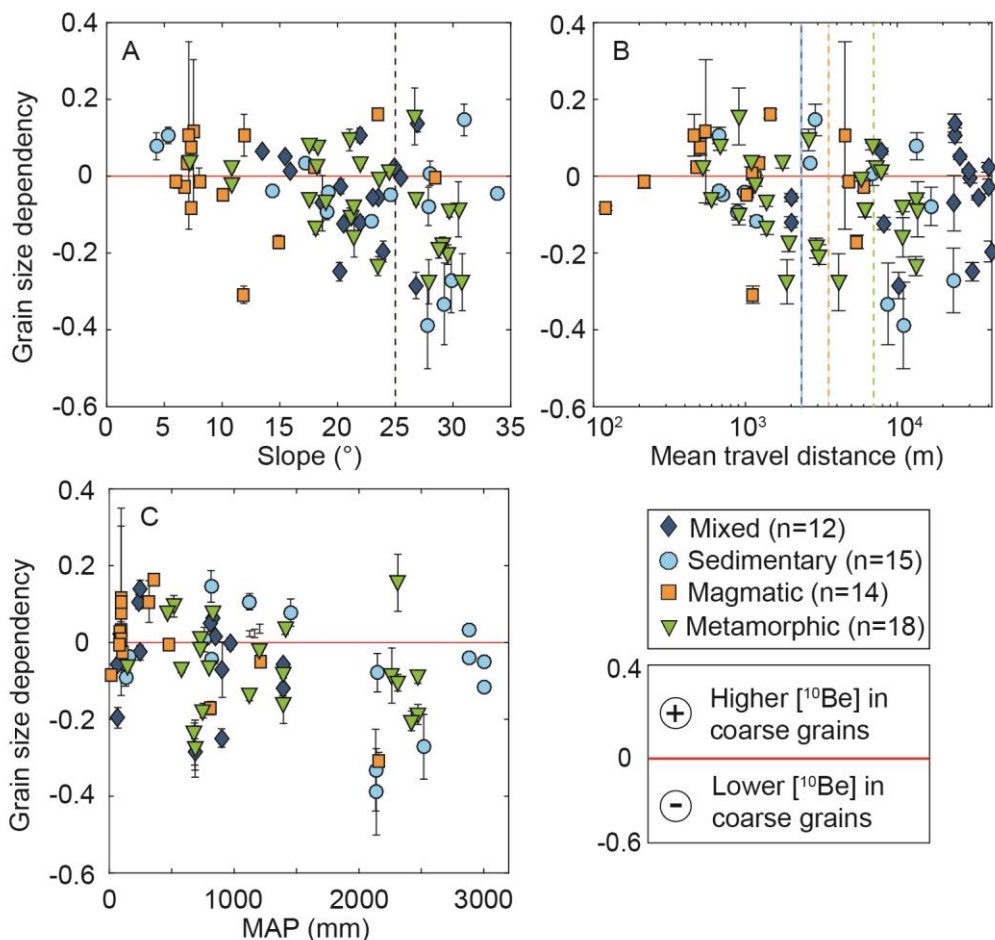

**Figure 8:** Grain size dependencies of all catchments in the global compilation (n=73), as a function of (A) mean basin slope, (B) mean travel distance and (C) mean annual precipitation (MAP). Coloured symbols depict lithological classes. Grain size dependencies are derived from the slope of a linear fit to the normalized $^{10}$Be concentrations and grain sizes from a sample set, as described in Figure 4. Dashed lines indicate the threshold hillslope (Figure A) and abrasion thresholds (Figure B) mentioned in the text. Global compilation statistics are provided in in Table S4 and Figure S4 of the data supplement.

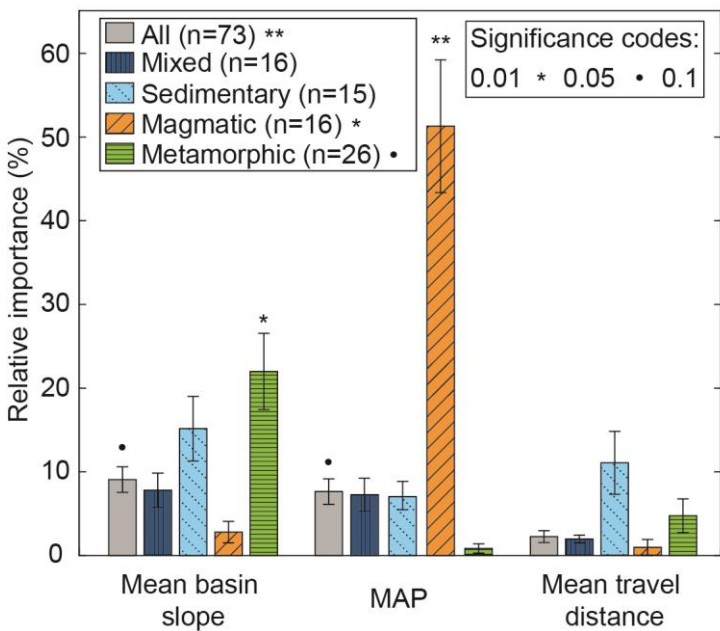

**Figure 9:** Relative importance of mean basin slope, MAP and mean travel distance to the multivariate linear regression $R^2$-value. Results are given for all lithologies combined (grey) and differentiated by lithology (colors). Multivariate linear regression results are provided in Table S5.

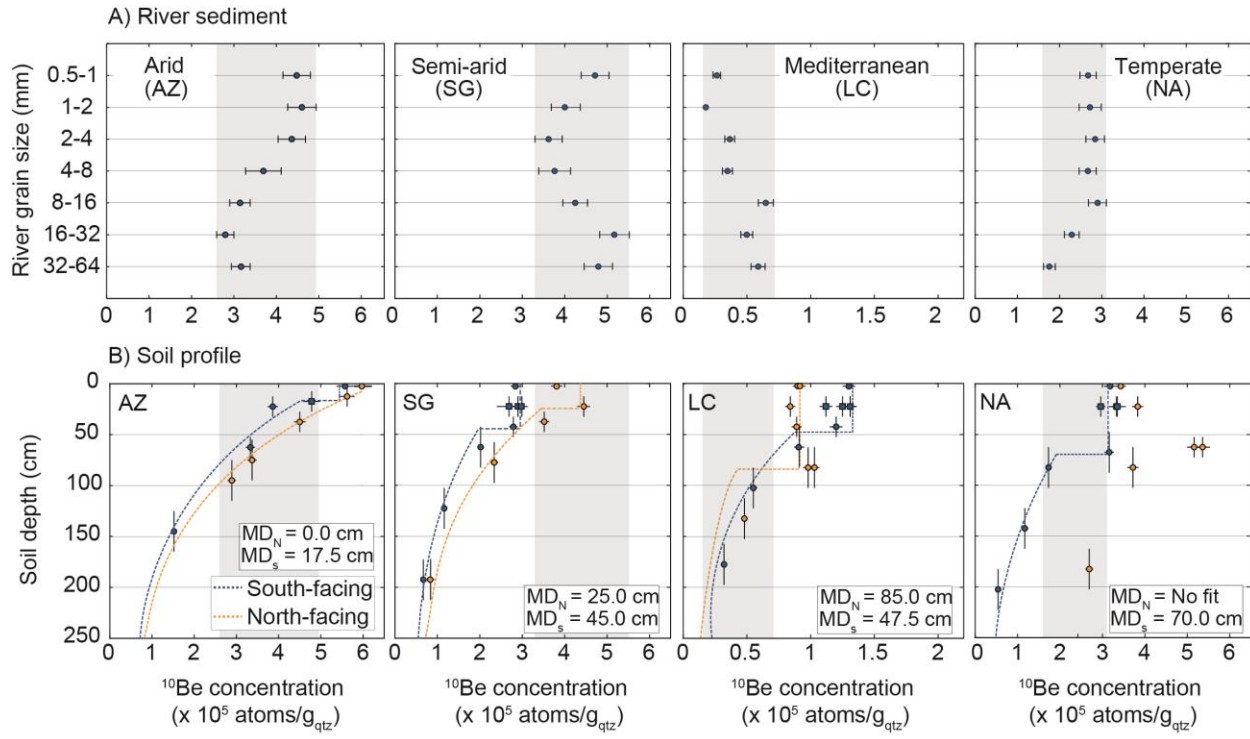


**Figure 10:** Comparison of $^{10}$Be concentrations measured in (A) river sediment, and (B) North and South-facing soil profiles (Schaller et al., 2018), from the same catchments in the Chilean Coastal Cordillera. $MD_N$ and $MD_S$ are the soil mixing depths of the North and South-facing hillslopes, respectively. Note the reduced x-axis range of the Mediterranean catchment (LC). Shaded areas show range of $^{10}$Be concentrations in river sediment for comparison with

soil profiles.