# Peer review of "Cosmogenic 10Be in river sediment: where grain size matters and why"

_Earth Surface Dynamics, 2018_

## Referee Comment (RC1) · Schmidt (Referee) · 21 Dec 2018

This paper combines analysis of new data from four catchments with a meta-analysis of previously published data to understand how and when in situ 10Be concentration is different in different grain size fractions from the same site. In line with previous, more local studies on this topic, the authors find that in areas with large deep seated landslides, grain size differences can be significant. They also identify sediment travel time and higher rainfall rates as reasons for increased differences in concentrations for different grain size fractions.

Overall I found the paper to be well written and thoughtfully analyzed. The authors figured out creative ways to quantitatively compare differences in grain size fractions

across a large dataset of previously published data. I have a few minor concerns that I think could probably easily be addressed and make the manuscript acceptable for publication.

1) The paper is relatively undercited. For example, a few times the use of in situ 10Be is mentioned for calculating background erosion rates and only two of the original three papers are cited. Given that all three groups did this work relatively independently and published at the same time, it is courteous to cite Brown et al. (1995), Granger et al. (1996), and Bierman and Steig (1996) any time the technique is brought up as a way to get long-term background erosion rates. All papers address the same assumptions underlying the technique. Another example is that papers by Gonzalez et al. (2016) in Geomorphology (one on Brazil, one on Panama) address the issue of grain size dependency in in situ 10Be for tropical locations. They conclude that differences are significant in tropical locations with deep seated landslides. Engaging with this literature would be a good idea (and this should happen at several points in the manuscript since it is quite relevant prior work). A third example is on line 60 on page 3 when the authors suggest that the larger analysis area makes it harder to distinguish trends due to increases in other variations. Portenga and Bierman suggest this exact point with their GSA Today paper and should probably be cited. Likewise, on line 318, I think it would be a good idea to cite Larsen's work on threshold hillslopes and erosion.

2) I wonder if the authors are overselling their results. I see a lot of points that overlap with the grey mean area on figure 4. Are there statistical tests that show that these differences are statistically significant? Is it because the difference between the 0.5-1 mm fraction is so different from the coarser fractions that suggests that there may be a problem? I didn't get that clearly from the figure.

3) I found the figures really confusing. Many of the figures are quite complex and have lots of data packed into them. The captions are short and don't really connect to how the figures were created, particularly for the ones with quote abstract scales (like normalized 10Be concentration). Clearly the authors have thought extensively

about how to display the data, but it is important to make sure that readers also have the benefit of those hours of thought. I think that some more clarification and a bit of improved plotting could help the figures significantly. Some kind of a summary figure showing differences across grain sizes for the meta analysis would make it much easier to interpret.

4) Pretty minor, but I'd like to see more connection back to the soil pits in the interpretation of the data. For example, in the paragraph starting on line 300, you talk about processes going on, and you have the soil pit figure, but you don't explicitly connect to the soil pits in the discussion.

5) Another minor thing - I am concerned that discussion of data interpretation but no input into writing doesn't really merit authorship. It seems like all authors should at least contribute to editing the ms before it gets published. I expect that they all did, but this should be noted in the contributions.

---

## Referee Comment (RC2) · Anonymous Referee #2 · 6 Jan 2019

Summary and general comments: This work provides an empirical analysis of variability in 10Be concentrations across detrital sediment sizes, using a case study from Chile and a global compilation of previous work. The authors assess the relative importance of mechanisms that likely contribute to this variability, including slope, travel distance, lithology, and precipitation. The results have implications for understanding sediment production on hillslopes, and provide useful information for study design in landscapes that may be susceptible to 10Be "grain size dependency" (i.e. systematic variation in 10Be concentrations with sediment size). This contribution is significant and relevant to a broad range of surface process and landscape evolution literature. It's great to see an empirical treatment of potential for bias in 10Be studies and the mechanisms that control 10Be across grain sizes, and I found this paper to be both timely and interesting.

I agree with the authors that their results suggest slope and travel distance are primary factors in grain size dependence (aside from the potentially larger role of lithology), and the fact that the effects of precipitation are measurable only at extremes (arid and humid) is quite interesting. I like the way the authors frame their synthesis in terms of thresholds, which is an intuitive and useful approach and is supported by the data. I found the figures to be mostly helpful, with the exception of Figure 6 (see more detailed comments below). The figure captions could be expanded a bit to help readers interpret the figures.

The main weaknesses I see are as follows (see specific comments below for more details):

1. The introduction and literature review needs a bit more attention, both in terms of clarity and including a broader range of relevant previous work.

2. The Chilean case study claims to isolate precipitation as a controlling factor on 10Be grain size dependency, but catchment area and slope vary significantly across these sites – this should be addressed in the discussion.

3. I'd love to see the effect of lithology explicitly teased out of the state-factor analysis (Figure 9 and associated text). The relative importance of MAP, slope, and travel distance is less meaningful without first quantifying how much of the variability might be attributed to lithologic controls. This should be relatively simple to do based on the current analysis. I think that the (potentially larger) role of lithology still needs additional work, and the details are certainly beyond the scope here – the discussion/conclusions could stress this point as a call to action from the community.

I also have questions about the normalized grain size approach and the consideration (or lack thereof) of changes in 10Be production rate across catchments. These are outlined in my specific comments below.

Overall, I found this paper to be reasonably well-written and easy to read. The analysis

and interpretations are thoughtful and well-reasoned, and the results are both interesting and novel. With some relatively small changes, I'd be happy to see it go forward.

Specific comments:

Section 2.1: There's a ton of information here, and there's a lot of relevant literature to point to. This section is essentially a literature review, but it leaves out some key papers – more references are needed in general, and a bit more care in the way the references are cited would be helpful. For instance, the first reference (Sklar et al., 2017) addresses the breadth of the first sentence, but the placement of the reference makes it seem like the paper is about mineralogy – there are other (earlier) papers that would be better references for this specific role in influencing sediment size. Re: weathering and climate, work by Riebe et al. (Sierra Nevada) and Dixon et al. (NZ Southern Alps) should probably also be included (unless line 79 refers only to the dependence of grain size on climate, rather than the dependence of weathering on climate – the text should be clarified on this point regardless).

In general, it's not clear if the literature reviewed here deals only with grain size variability, or with the variability in erosion, weathering, lithology, climate, etc. that influence hillslope processes (and therefore grain size), even in studies where grain size is not explicitly addressed. Section 2.1 should probably be expanded a bit to more clearly explain how these mechanisms relate to grain size. It also needs more citations to acknowledge the body of literature behind each of these topics.

The metrics identified at line 120 as the controlling factors on grain size and 10Be dependence aren't introduced in the previous section explicitly. This section could be much improved either 1) by taking a more linear path to get to these factors (which would require a restructure of the introduction) or 2) by adding a bit more explanation here re: why these particular factors are important (e.g. provide some specific mechanistic examples). They're not exactly simple state factors, there are a lot of complicated interactions. Slope should influence erosion rate and susceptibility to landslides. Pre-

СЗ

cipitation should also influence erosion rates and weathering intensity (though there's literature on both sides of this argument, which is not acknowledged here). I'm not concerned about your choice of metrics – these are great things to quantify – but a bit more massaging of the text would help clarify exactly what you hope to learn in designing the study this way.

The case study in the Chilean Cordillera claims to isolate precipitation as a controlling factor on 10Be variability across grain sizes. While these sites span relatively similar lithology and tectonic uplift, catchment area and slope vary significantly across these sites. These factors should also be considered in the analysis and discussion.

Figure 6: I find this figure difficult to read. There is some benefit to having all 4 metrics displayed at once, but I think it would be preferable to split this information into more figures/panels. It's just too much to take in, the trends and details get lost. Maybe move this figure to the supplemental material, and provide a clearer set of plots. I found figure S4 to be really useful because everything was split out into separate plots. Providing separate plots with the absolute grain size and normalized grain size would be useful, as I don't find the normalized grain size to be intuitively as useful – maybe that actually means moving Fig. S4 into the main text?

The normalized grain size is calculated using the arithmetic mean of grain sizes from the same catchment, but what does that mean (average) really mean (signify)? Is it the average of grain sizes in which 10Be was measured, or the average grain size present on the streambed? If no pebble counts were reported (as I'm sure they weren't for all of these studies), is the average of sizes in which 10Be was measured really all that useful? I realize you're trying to find a way to compare across a huge swath of literature, with highly variable sampling approaches, and you need a way to compare across studies. If the goal of the compilation is to understand how 10Be varies across sediment size, the actual (rather than normalized) sediment sizes are potentially quite important, and that information gets lost in Fig. 6. Results from different lithologies (starting at line 265) – to me, these observations suggest that lithology is a fundamental control on grain size dependency. If you're going to do multi-variate analysis to attribute grain size variability to each factor, can you tease apart the variability attributable to lithology first, and then discuss trends attributable to other factors? Ideally you'd do a multi-variate analysis for all 4 factors, but lithology isn't a continuous dataset, so this isn't really possible – teasing out the affects of lithology first would at least give you an idea of the relative importance of each factor including lithology.

Figure S5 suggests (or at least asserts – providing a calculation to back it up would be useful) that the altitudinal variation in 10Be production is not sufficient to explain the positive trend in 10Be with grain size. What about the other catchments? If coarse grains originated only at low elevations (i.e. were not transported from upper parts of the catchment), could that explain negative grain size dependency? Could this be a sediment transport story, rather than a landslide/depth shielding story? By ignoring the spatial variation in 10Be production, you're essentially assuming that sediment originates from all elevations – this assumption may not always be valid for all grain sizes, and it should probably be stated somewhere in the text.

Technical comments:

Line 44: modeling studies (Lukens et al., 2016) help constrain how big this bias could be (under- or overestimating erosion rates by a factor of 3 or more) – this might be worth mentioning here as context for the potential scope of the problem.

Line 107: "Furthermore, fluvial processes can affect grain size fractions in a way that not all parts of a catchment are equally represented at a given sample location" – this is certainly true, but needs to be more clearly explained and certainly needs references. (You do this at the end of the paragraph, maybe this sentence just needs to move.)

Line 113: "grain size" here refers to mineral grains, yes? Be careful/specific when discussing mineral grains vs clast sizes. These are different problems to consider, and

arise for different reasons (lithologic controls vs. weathering/geomorphic process/etc.). I've run into a fair bit of confusion from readers/reviewers in my own work for just this reason, and the only advice I can give is that clarity and consistency of language around this distinction is paramount – I'd suggest changing "grain size" here to "mineral size".

Line 118: misplaced comma towards the end of the line (should go after "depth")

Line 123: "process" should be plural

Line 246: "Uncertainties in 10Be concentrations tend to be higher for samples from steeper hillslope angles (>10°), which is related to generally lower concentrations, i.e., higher denudation rates." This sentence is confusing. On my first read I thought you were suggesting that hillslope angle controlled 10Be uncertainties. Too many ideas in one sentence, break it up for the sake of clarity. Steeper hillslopes are eroding faster, which means they have lower 10Be concentrations. Uncertainties are larger for very low 10Be/9Be ratios.

Line 248: "nearly similar" is redundant, nix the "nearly"

Line 277: "Only in the arid. . ." Awkward sentence structure – consider flipping it around (Trends in 10Be only exist in the arid and mediterranean catchments)

Line 287: missing comma before figure reference

Line 289: Are the soil pit 10Be measurements also from Schaller et al. 2008? If so, cite them here. Referencing Figure 8 here would also be appropriate.

Line 317: "threshold slopes. . . where hillslopes cannot get any steeper" It's a nitpicky point, but I'd quibble with this language. Conceptually, yes, slopes steeper than the threshold "shouldn't" exist, but they certainly do at local scales. Your plots have slopes steeper than the threshold, even at basin-averaged scales, so your own data attest to the fact that hillslopes CAN get steeper.

Line 356: "We thus think" = awkward

Line 357: missing "the" before "mixed soil layer"

Line 370: don't start a sentence with "And"

---

## Referee Comment (RC3) · Anonymous Referee #3 · 9 Jan 2019

This is an interesting and timely study presented by van Dongen et al., assessing the influence of the grain size used in detrital 10Be sampling and how it may bias measured concentrations and subsequently estimated catchment averaged denudation rates. The authors present new CRN data from a series of catchments in the Chilean Coastal Cordillera which span across a notable climatic gradient, to test the effect of precipitation on 10Be grain size dependence. They combine this with an analaysis of other metrics in these catchments (e.g. hillslope angle, lithology, abrasion) which are likely to produce grain size fractions with variable 10Be concentrations. Finally, a similar analysis is performed on a global dataset of 10Be samples and reported grain size fractions to test whether grain size dependencies exist. Overall, I found the paper interesting and relatively well written. I think there are some important messages

concerning the conditions under which grain size dependency may bias 10Be sampling which can be drawn from the manuscript. With some clarification and moderate changes, I would support publication of the manuscript.

**General comments:**

One of my first thoughts on reading this manuscript is the overlap with that of Binnie et al. (2007) Geology, although there is no reference to this study. One of the main finding from this earlier study was that denudation rate and slope gradient are broadly linear up to threshold hillslope gradients of  $\sim$ 30 degrees. Beyond this, denudation rates are much more variable because of a transition from transport-limited to detachment-limited denudation processes – i.e. steepening the hillslope beyond this has limited effect on catchment averaged denudation rate. In general, I felt there were a number of key references missing in the first few sections of this manuscript. I think it needs to be made clearer how the findings from this study are new/different from previous work.

Production rates: In terms of CRN production rate varying measured concentrations – I'm not sure what the total relief across the Cordillera catchments are (could you possibly add this into Table 1?) or the CRN production rate that was used (please also add this in somewhere) for each catchment but it seems unlikely that this would have a significant impact unless there is some considerable relief?

Lithology: Just a quick query about quartz content of lithologies – I'm presuming most of these lithologies are fairly quartz rich such that this shouldn't bias any results (I'm thinking mostly about the mixed category)? Line 130 states some variations in mineralogy exist – do you have any maps/indication as to whether this is significant in terms of biasing quartz distribution across the catchment?

Catchment size: I'm not entirely convinced by the interpretation on L326. Looking at Figure 7b, there does appear to be some increase in negative grain size dependency in larger catchments but only really in sedimentary lithologies. Is the scale correct on the horizontal axis in Figure 7b (I thought only catchments <5000 km2 were considered)?

In these larger catchments, I'd expect that these sedimentary rocks would abrade more quickly into finer grain size fractions, especially given these greater travel distances – perhaps these coarser and lower concentration grains are actually more locally produced (lower production rate if from lower elevations too possibly?). In the sections following Line 332, there is a counter argument that in really large catchments (the exact size would depend on lithology/abrasion thresholds I presume) the effects of grain size dependency are likely to be less, as the majority of material should be abraded into sand? I think this comment rests upon whether the horizontal scale in Figure 7b is correct or not. It might be helpful to define what you consider as a large catchment here (>1,000 km2 etc.)?

Lupker et al. (2012) considered multiple grain size fractions in samples taken from a number of Himalayan catchments at the mountain front and found no systematic trend or differences in 10Be concentrations as a function of sand grain size (which makes up the majority of the sediment load). With increasing catchment area, one would expect the concentrations measured in the sand fraction to be more representative of the total catchment. As catchments get larger, there are also likely to be different erosional processes operating within in which may influence 10Be concentrations (see Dingle et al., 2018). For example glacial shielding (which will offset any difference in production rate as a function of elevation), glacial erosion, sediment recycling and 'hotspots' of erosion which may be driven by spatial variations in climate which can occur across sufficiently large catchments (e.g. localised storms), or parts of the catchment which undergo higher rates of rock uplift and are more susceptible to landsliding. There are then also issues relating to temporary storage (even just within the channel itself, or within large landslides) within increasingly larger catchments. I think you touch upon this in Line 59.

Temporal effects: One of the key aspects I feel this manuscript currently overlooks is a discussion on how representative the Cordillera samples are. These catchments are small (<10 km) and experience landsliding – how likely is it that these samples

СЗ

are influenced by the stochastic nature of sediment delivery from these landslides (e.g. Niemi et al., 2005 – I noticed that this paper wasn't referenced anywhere). Do you have truly 'representative' samples and how stable are these concentrations in different grain sizes in time? Is material generated by these deposits likely to be well mixed into the suspended/bed material load (especially given such short transport distances), or is it likely to overwhelm the catchment-averaged signal depending on factors such as the time since the last event/time since significant mobilisation of landslide material.

Grain size: It took me a while to get my head around what the normalized grain size statistics actually represent, especially given the ranges may have varied between the studies looked at (e.g. where only >2 mm was stated, values were forced into 2-4 mm). If I have this right, the grain size fractions presented in each study could influence the normalized grain size you calculate if these fractions were inconsistent between studies? It would be really nice to see metrics plotted against absolute grain size (maybe somewhere in the SI) given you have this information available. Another point which I think Reviewer 2 has also commented on – what is a representative grain size of a catchment (thinking about Line 28-29 in the abstract in particular), and is this what is being sampled on the river bed (e.g. Figure S2 shows that some of the catchments capture more of the CDF than others)? Interesting that your bedload GS in Chile is much more bimodal than in either Italy or the SGM datasets (Figure 1).

Specific comments:

Line 52 – Which other studies? Add some references

Line 87-94 – I found this paragraph a little wordy. "when the transport capacity of the water is too low" - too low for what? Please be more specific.

Line 95-97 - "Any process that transports different grain sizes, from areas in a catchment with contrasting....". You could also say the same for grains of the same size from different parts of the catchment. I feel that this paragraph could do with a little more work generally. For example, in Line 107 you discuss variations in 10Be concentrations in soil as a function of whether the landscape is eroding quickly or not. There is the argument that in more rapidly eroding landscape you would only expect larger variations in concentrations (due to removal of material from depths greater than the attenuation length) if a concentration profile is fully developed. In rapidly eroding landscapes, you may just end up with upper layers characterized by relatively (uniformly) low concentration material? The jump to the metrics you propose to look at in L117 onward feels quite big. It would be nice to see a clearer build up to this in the paragraph beforehand (Line 70 onwards) so that it is obvious why these metrics have been chosen.

Line 174 - what happens if you remove these studies (stated only as >2 mm) from your statistics? I appreciate this may remove a large number of points but might be interesting to see.

Line 246 – Are these uncertainties relating to error/uncertainty in the laboratory measurements or variability in measured concentrations? Either way, uncertainty and variability are different so please clarify!

Line 250 – While no pattern in MAP, I wonder whether the frequency of large storms is a factor that is likely to be important? Is it fair to assume that all sediment generated by landsliding in these catchments is immediately evacuated from the catchment and there is no preferential mobility of coarser/finer material (i.e. it might take a large storm to mobilise the coarsest material which may only happen a few days of the year?).

Line 254 – Is 54.8% really 'significant'? In general I found some of the statistics a little weak and definitions of coarse/fine not fully stated. For example, on line 234 and Figure 4, you state that only the AZ and LC catchment show consistent trends between 10Be and grain size. When I look at Figure 4 I see a lot of scatter/variability!

Line 264 - 'Partly accentuated and partly muted' - this is a very confusing sentence!

Line 278 – 'In both catchments the 10Be concentrations of river sediment correspond to concentrations measured deeper within soil profiles' – looking at AZ and LC in Figure

8 it looks like the river concentrations correspond to the concentrations measured in the upper 1m of the soil profile (AZ), not material from greater depths. In LC, it looks like all of the river grain size fractions are consistent with concentrations measured below 1m in the soil puts, suggesting no grain size dependence. Instead this seems to suggest that all of the sampled sediment is overwhelmed by material excavated from depth? Could the fact that the river sediment concentrations are lower than those in the soil pit (line 288) simply reflect the fact that the concentrations measured in the soil pits are not representative of the entire catchment?

Line 300 – I find this sentence undermines the study slightly...maybe consider rephrasing 'our new samples from the Chilean Coastal Cordillera demonstrate minor variations in 10Be concentrations'.

Table 2 – what is the superscript b referring to in the last column title?

Figure 2 - Line 707 - 'constant' or maybe uniform?

Figure 6 - I suspect this is one of the key figures for the paper but find it difficult to follow. There is a lot of information in there.

Figure 9 – 'results given for all lithologies combine' – should be 'combined'

---

## Author Comment (AC1) · 11 Mar 2019

We thank A. Schmidt, and two anonymous reviewers for their constructive feedback on our manuscript. We believe their reviews allowed us to improve the manuscript significantly. The authors response was uploaded in the form of a PDF-supplement.

Please also note the supplement to this comment:
https://www.earth-surf-dynam-discuss.net/esurf-2018-83/esurf-2018-83-AC1-supplement.pdf

---

## Author Response (AR2)

**Authors response to reviews on van Dongen et al. - Cosmogenic 10Be in river sediment: where grain size matters and why**

We thank A. Schmidt, and two anonymous reviewers for their constructive feedback on our manuscript. We believe their reviews allowed us to improve the manuscript significantly. Major revisions include:

- 1. We changed normalized grain size classes to phi-classified grain size classes, which resulted in slight changes in statistical results; however, conclusions remained generally the same.
- 2. We included 13 additional recently published catchments (Tofelde et al., 2018).
- 3. We improved the literature review as suggested by the reviewers.

This document is built up as following. First, we cite the comment from the reviewers in black Roman font. We numbered the reviewer comments first starting with the comment of reviewer 1, 2 or 3, following with the number of the comment after the point (1.1 is comment number 1, of reviewer 1). Our response is provided in blue Roman font and revisions are given in *blue italic* font, in which we *underlined* the corrections.

This document contains the following structure:

- 1. Comment on review of referee A. Schmidt
- 2. Comment on review of Anonymous Referee #2
- 3. Comment on review of Anonymous Referee #3
- 4. Revised manuscript with track changes

**1. Comments on review of Referee Schmidt:**

This paper combines analysis of new data from four catchments with a meta-analysis of previously published data to understand how and when in situ 10Be concentration is different in different grain size fractions from the same site. In line with previous, more local studies on this topic, the authors find that in areas with large deep seated landslides, grain size differences can be significant. They also identify sediment travel time and higher rainfall rates as reasons for increased differences in concentrations for different grain size fractions. Overall I found the paper to be well written and thoughtfully analyzed. The authors figured out creative ways to quantitatively compare differences in grain size fractions across a large dataset of previously published data. I have a few minor concerns that I think could probably easily be addressed and make the manuscript acceptable for publication. Reviewer Comment 1.1: The paper is relatively undercited. For example, a few times the use of in situ 10Be is mentioned for calculating background erosion rates and only two of the original three papers are cited. Given that all three groups did this work relatively independently and published at the same time, it is courteous to cite Brown et al. (1995), Granger et al. (1996), and Bierman and Steig (1996) any time the technique is brought up as a way to get long-term background erosion rates. All papers address the same assumptions underlying the technique.

Authors response: Thanks for your alertness, we have included Brown et al., 1995 for a complete documentation.

**Revisions made:**

*Line 33: Catchment-average denudation rates are commonly estimated with in situ-produced cosmogenic* 10*Be concentrations in river sediment (Bierman and Steig, 1996a; Brown et al., 1995; Granger et al., 1996).*

Line 36: Its concentration records the time minerals were exposed to cosmic radiation, which is inversely proportional to denudation rates over time scales of  $10^2$ - $10^5$  years (Bierman and Steig, 1996b; Brown et al., 1995; Granger et al., 1996; Lal, 1991).

*Line 40: The sand fraction provides a representative catchment-average denudation rate only if it is spatially and temporally representative for all erosion sources within the catchment (Bierman and Steig, 1996b; von Blanckenburg, 2005; Brown et al., 1995; Granger et al., 1996; Neilson et al., 2017; Sosa Gonzalez et al., 2017; Willenbring et al., 2013).*

Reviewer Comment 1.2: Another example is that papers by Gonzalez et al. (2016) in Geomorphology (one on Brazil, one on Panama) address the issue of grain size dependency in in situ 10Be for tropical locations. They conclude that differences are significant in tropical locations with deep seated landslides. Engaging with this literature would be a good idea.

Authors response: We have included Sosa Gonzalez et al., 2016a and Sosa Gonzalez et al., 2016b, at 3 positions in the revised manuscript.

**Revisions made:**

Line 47: Some studies inferred that lower 10Be concentrations in coarse grains are caused by deep-seated erosion processes, such as landslides, which excavate material from greater depth where 10Be concentrations are lower (e.g. Aguilar et al., 2014; Belmont et al., 2007; Binnie et al., 2007; Brown et al., 1995; Puchol et al., 2014; Sosa Gonzalez et al., 2016a, 2016b; Tofelde et al., 2018; West et al., 2014).

Line 124: Because 10Be production rates decrease exponentially with depth (Gosse and Phillips, 2001), hillslope sediment that is excavated over a larger depth interval by landslides will obtain a larger variation in 10Be concentrations than sediment transported by diffusive processes near the surface (Aguilar et al., 2014; Belmont et al., 2007; Binnie et al., 2007; Brown et al., 1995; Puchol et al., 2014; Sosa Gonzalez et al., 2016a, 2016b; Tofelde et al., 2018; West et al., 2014).

Line 401: This conforms with previous studies that also found negative grain size dependencies which emerged from a transition of transport-limited to detachment-limited erosion processes and, therefore, deep-seated erosion processes (Binnie et al., 2007; Brown et al., 1995; Lukens et al., 2016; Reinhardt et al., 2007; Sosa Gonzalez et al., 2016a, 2016b; Tofelde et al., 2018).

Reviewer Comment 1.3: A third example is on line 60 on page 3 when the authors suggest that the larger analysis area makes it harder to distinguish trends due to increases in other variations. Portenga and Bierman suggest this exact point with their GSA Today paper and should probably be cited.

Authors response: Good point, we included 2 citations discussing this issue.

**Revisions made:**

Line 62: A reason may be that as catchment size increases, more complexity in controlling factors is added that may blur potential trends in the data (Carretier et al., 2015; Portenga and Bierman, 2011).

Reviewer Comment 1.4: Likewise, on line 318, I think it would be a good idea to cite Larsen's work on threshold hillslopes and erosion.

Authors response: Included.

*Revisions made:*

*Line 397: In these catchments, many hillslopes have likely reached the threshold hillslope angle of ~25-30°, at which denudation rates are dominated by the frequency of landslides (Larsen and Montgomery, 2012; Montgomery and Brandon, 2002; Ouimet et al., 2009).*

Reviewer Comment 1.5: I wonder if the authors are overselling their results. I see a lot of points that overlap with the grey mean area on figure 4. Are there statistical tests that show that these differences are statistically significant? Is it because the difference between the 0.5-1 mm fraction is so different from the coarser fractions that suggests that there may be a problem? I didn't get that clearly from the figure.

Authors response: We understand the confusion and agree that the figure can be clarified. The grey shaded area showed the total variability of 10Be concentrations in the catchment, based on all grain size classes. It is, therefore, not related to the significance of the individual samples. When error bars ( $2\sigma$ -analytical errors) on the 10Be concentrations do not overlap, the samples are significantly different. To prevent confusion, we removed the grey shaded area and discussed the catchment variability in the text only. However, we think it is incorrect to apply a statistical test to proof significant differences between grain sizes, because each grain size class exist of one single sample and not a normal distribution of multiple samples.

Reviewer Comment 1.6: I found the figures really confusing. Many of the figures are quite complex and have lots of data packed into them. The captions are short and don't really connect to how the figures were created, particularly for the ones with quote abstract scales (like normalized 10Be concentration). Clearly the authors have thought extensively about how to display the data, but it is important to make sure that readers also have the benefit of those hours of thought. I think that some more clarification and a bit of improved plotting could help the figures significantly. Some kind of a summary figure showing differences across grain sizes for the meta analysis would make it much easier to interpret.

Authors response: We regret that the figures and figure captions were not sufficiently clear. In the revised version, we aimed to clarify figures and their captions and also modified the main text in many places. The main changes to the figures are as follows: 1) We introduced a new figure (Fig. 6) in the methodology which explains how the grain size dependency values were derived and what they represent. 2) We used absolute phi-grain size classes, which are not normalized. 3) We removed the TH and AT abbreviations and shaded areas from Fig. 7 and explained the dashed lines in the figure caption. 4) We emphasized in the text that Fig. 7 for individual lithologies can be found in the data supplement. 5) We checked all figure captions and added explanations in several places. We hope these changes improve the figures and their readability.

Reviewer Comment 1.7: Pretty minor, but I'd like to see more connection back to the soil pits in the interpretation of the data. For example, in the paragraph starting on line 300, you talk about processes going on, and you have the soil pit figure, but you don't explicitly connect to the soil pits in the discussion.

Authors response: To improve the connection to the soil pits we added a sentence about the differences between catchment-average and the soil pit production rates and compared catchment-average 10Be concentrations to soil pit 10Be concentrations to infer excavation depths of erosion processes. We also included the locations of the soil pits in the study area figure (Figure 2).

**Revisions made:**

Line 343: Because the difference between 10Be production rates of the catchment on average and at the soil profiles is small (<10%), we can compare measured 10Be concentrations directly.

Reviewer Comment 1.8: Another minor thing - I am concerned that discussion of data interpretation but no input into writing doesn't really merit authorship. It seems like all authors should at least contribute to editing the ms before it gets published. I expect that they all did, but this should be noted in the contributions.

Authors response: You are right, we included this to the author's contributions paragraph.

*Revisions made: Line 503: R. van Dongen prepared the manuscript with contributions and edits from all co- authors.*

**2. Comments on review of Anonymous Referee #2:**

Summary and general comments: This work provides an empirical analysis of variability in 10Be concentrations across detrital sediment sizes, using a case study from Chile and a global compilation of previous work. The authors assess the relative importance of mechanisms that likely contribute to this variability, including slope, travel distance, lithology, and precipitation. The results have implications for understanding sediment production on hillslopes, and provide useful information for study design in landscapes that may be susceptible to 10Be "grain size dependency" (i.e. systematic variation in 10Be concentrations with sediment size). This contribution is significant and relevant to a broad range of surface process and landscape evolution literature. It's great to see an empirical treatment of potential for bias in 10Be studies and the mechanisms that control 10Be across grain sizes, and I found this paper to be both timely and interesting.

I agree with the authors that their results suggest slope and travel distance are primary factors in grain size dependence (aside from the potentially larger role of lithology), and the fact that the effects of precipitation are measurable only at extremes (arid and humid) is quite interesting. I like the way the authors frame their synthesis in terms of thresholds, which is an intuitive and useful approach and is supported by the data. I found the figures to be mostly helpful, with the exception of Figure 6 (see more detailed comments below). The figure captions could be expanded a bit to help readers interpret the figures.

**2.1Main weaknesses:**

Reviewer Comment 2.1: The figure captions could be expanded a bit to help readers interpret the figures.

Authors response: We expanded the figure captions and improved our figures 6 and 7 (figure 7 and 8 in revised manuscript); see our detailed response to Reviewer comment 1.6.

Reviewer Comment 2.2: The introduction and literature review needs a bit more attention, both in terms of clarity and including a broader range of relevant previous work.

Authors response: We improved the literature review and included more references, see a detailed description below (Reviewer comment 2.6) and also our response to a similar comment by Reviewer 1 (Reviewer comments 1.1-1.4).

Reviewer Comment 2.3: The Chilean case study claims to isolate precipitation as a controlling factor on 10Be grain size dependency, but catchment area and slope vary significantly across these sites – this should be addressed in the discussion.

**Authors response: See our response below (Reviewer Comment 2.8).**

Reviewer Comment 2.4: I'd love to see the effect of lithology explicitly teased out of the state-factor analysis (Figure 9 and associated text). The relative importance of MAP, slope, and travel distance is less meaningful without first quantifying how much of the variability might be attributed to lithologic controls. This should be relatively simple to do based on the current analysis. I think that the (potentially larger) role of lithology still needs additional work, and the details are certainly beyond the scope here – the discussion/conclusions could stress this point as a call to action from the community.

Authors response: See our response to the more detailed comment below (Reviewer comment 2.11).

**2.2Specific comments:**

Reviewer Comment 2.5: Section 2.1: There's a ton of information here, and there's a lot of relevant literature to point to. This section is essentially a literature review, but it leaves out some key papers – more references are needed in general, and a bit more care in the way the references are cited would be helpful. For instance, the first reference (Sklar et al., 2017) addresses the breadth of the first sentence, but the placement of the reference makes it seem like the paper is about mineralogy – there are other (earlier) papers that would be better references for this specific role in influencing sediment size. Re: weathering and climate, work by Riebe et al. (Sierra Nevada) and Dixon et al. (NZ Southern Alps) should probably also be included (unless line 79 refers only to the dependence of grain size on climate, rather than the dependence of weathering on climate – the text should be clarified on this point regardless).

Authors response: We see your point. We expanded the review section, included more references (among others: Dixon et al., 2016 and Riebe et al., 2004) and moved the first reference (Sklar et al., 2017) to the general sentence that introduces the controlling factors.

**Revisions made:**

*Line 77: Chemical weathering at the hillslope converts bedrock into sediment of different grain sizes at a rate that is controlled by the properties of the parent rock, the climatic regime, and denudation rates at the surface (Sklar et al., 2017).*

Line 82: As chemical weathering rates are set primarily by the flux of water flowing through the regolith and, to some degree, also by temperature (Lasaga et al., 1994; Maher, 2010; White et al., 1999), there is a strong dependency of weathering and sediment production on the climatic regime (Dixon et al., 2016; Riebe et al., 2004).

Reviewer Comment 2.6: In general, it's not clear if the literature reviewed here deals only with grain size variability, or with the variability in erosion, weathering, lithology, climate, etc. that influence hillslope processes (and therefore grain size), even in studies where grain size is not explicitly addressed. Section 2.1 should probably be expanded a bit to more clearly explain how these mechanisms relate to grain size. It also needs more citations to acknowledge the body of literature behind each of these topics.

Authors response: We acknowledge the criticism and made the link to grain sizes clearer. Furthermore, we added additional references to several sections in the literature review.

**Revisions made:**

Line 77: Chemical weathering at the hillslope converts bedrock into sediment of different grain sizes at a rate that is controlled by the properties of the parent rock, the climatic regime, and denudation rates at the surface (Sklar et al., 2017).

*Line 79: The parent rock mineralogy sets rock dissolution rates and constrains the minimum size of individual minerals.*

Line 82 As chemical weathering rates are set by the temperature and the flux of water flowing through the regolith (Lasaga et al., 1994; Maher, 2010), there is a strong dependency of weathering and sediment production on the climatic regime (Dixon et al., 2016; Riebe et al., 2004).

Line 85: The presence of biota in humid climates can enhance the breakdown of rock fragments because microbes play an important role in chemical weathering processes, and growing roots can fracture bedrock (Drever, 1994; Ehrlich, 1998; Gabet and Mudd, 2010; Roering et al., 2010).

Reviewer Comment 2.7: The metrics identified at line 120 as the controlling factors on grain size and 10Be dependence aren't introduced in the previous section explicitly. This section could be much improved either 1) by taking a more linear path to get to these factors (which would require a restructure of the introduction) or 2) by adding a bit more explanation here re: why these particular factors are important (e.g. provide some specific mechanistic examples). They're not exactly simple state factors, there are a lot of complicated interactions. Slope should influence erosion rate and susceptibility to landslides. Precipitation should also influence erosion rates and weathering intensity (though there's literature on both sides of this argument, which is not acknowledged here). I'm not concerned about your choice of metrics – these are great things to quantify – but a bit more massaging of the text would help clarify exactly what you hope to learn in designing the study this way.

Authors response: Thanks for pointing out that the aim of our study didn't get across well. Besides modifying the literature review, we added a final paragraph, in which we discuss which catchment attributes may be indicative of the processes identified in the literature review and how we tackle them in our study. This section thus provides a useful link between the literature review and the methods section.

**Revisions made:**

**Line 144:* **2.3** *Catchment attributes that potentially control grain size-dependent* 10*Be concentrations**

Based on the above review of processes that may influence grain size-dependent 10Be concentrations, we identified the catchment attributes mean basin slope, mean travel distance, mean annual precipitation (MAP), and lithology that we will focus on in our study. Here, we consider mean basin slope as a topographic catchment attribute that controls denudation rates and the scouring depth of diffusive or deep-seated erosion processes. We selected MAP because of its effect on both weathering rates and the scouring depth of erosion processes and lithology because it affects chemical weathering rates, the grain size of individual minerals and the susceptibility to hillslope failure. Finally, we selected mean travel distance of sediment as a metric for fluvial processes that are transport-dependent (e.g. abrasion and hydrodynamic sorting).

Reviewer Comment 2.8: The case study in the Chilean Cordillera claims to isolate precipitation as a controlling factor on 10Be variability across grain sizes. While these sites span relatively similar lithology and tectonic uplift, catchment area and slope vary significantly across these sites. These factors should also be considered in the analysis and discussion.

Authors response: We agree that the La Campana catchment has a different mean basin slope angle and total relief compared to the 3 other catchments. Yet, although mean basin slope and total relief values of the other 3 catchments vary, but the values are generally low (8-17°) and, compared to the global compilation, for example, rather minor. Differences in catchment area are more significant – especially AZ is relatively small (~0.04 km2) compared to the other catchments (~1-7 km2) and we now address this difference in the discussion. In short, we expect that the effect of abrasion on grain size is minor, as granitic rocks have a low breakdown rate and the mean travel distance is short.

**Revisions made:**

Line 371: We propose that the existing or missing trends in the arid (AZ), semi-arid (SG) and temperate (NA) catchments are mainly related to differences in precipitation and the excavation depth of the erosion processes. These catchments show minor variations in mean basin slope, hence we do not expect big differences in erosion processes due to changes in slope alone. Furthermore, the limited relief of these catchments excludes differences in 10Be production rates and local sediment sources to influence observed differences in 10Be concentrations.

*Line 376:* However, steeper hillslope angles and higher total relief may have overruled the effect of precipitation in the La Campana catchment. We do not expect a control related to the different catchment sizes in any of the catchments, because granitic rock have a low abrasion breakdown rate (Attal and Lavé, 2009) and the mean travel distances were small (<1 km).

Reviewer Comment 2.9: Figure 6: I find this figure difficult to read. There is some benefit to having all 4 metrics displayed at once, but I think it would be preferable to split this information into more figures/panels. It's just too much to take in, the trends and details get lost. Maybe move this figure to the supplemental material, and provide a clearer set of plots. I found figure S4 to be really useful because everything was split out into separate plots. Providing separate plots with the absolute grain size and normalized grain size would be useful, as I don't find the normalized grain size to be intuitively as useful – maybe that actually means moving Fig. S4 into the main text?

Authors response: We realized that figures 6 and 7 need clarification (figure 7 and 8 in revised manuscript). However, we are concerned that providing the figures separated by lithology in the main manuscript will overload the reader with a large number of figures. Instead, we additionally emphasize in the text that figures separated by lithology can be found in the data supplement. We also helped the reader in understanding these figures by providing an additional figure (Figure 4) that explains the concept of grain size trend. Finally, we discarded the normalized grain size and instead use phi-grain size classes, which essentially linearize the grain sizes and facilitate reading the trends in 10Be concentration with grain size. Please, see also our detailed response to Reviewer Comment 1.6.

**Revisions made:**

*Line 295: Figure 8 (and Figure S4, for plots separated by lithology) shows the grain size dependencies of individual catchments, resulting from the slope of a linear fit to the 10Be concentrations of all grain size classes (see methods section and Figure 4).*

Reviewer Comment 2.10: The normalized grain size is calculated using the arithmetic mean of grain sizes from the same catchment, but what does that mean (average) really mean (signify)? Is it the average of grain sizes in which 10Be was measured, or the average grain size present on the streambed? If no pebble counts were reported (as I'm sure they weren't for all of these studies), is the average of sizes in which 10Be was measured really all that useful? I realize you're trying to find a way to compare across a huge swath of literature, with highly variable sampling approaches, and you need a way to compare across studies. If the goal of the compilation is to understand how 10Be varies across sediment size, the actual (rather than normalized) sediment sizes are potentially quite important, and that information gets lost in Fig. 6.

Authors response: That is a good point. Normalizing the grain sizes by the mean value of all grain sizes in each catchment results in different normalization values for each catchment.

Our main reason to normalize grain size classes was driven by our comparison of the slope of a linear model fit to the grain size-10Be data. Differences in measured grain sizes yield different slope values, even if the differences in normalized 10Be were the same. In the revised version, we circumvented this problem without normalizing the data by log-transforming grain sizes to phi-grain size classes and changed all figures, statistical tests, and calculations accordingly.

**Revisions made: Phi-grain size ranges are plotted on the x-axis of all figures related to the global compilation (Figure 5, 6, 7 and 8). Table S4 and S5 include the new statistical results.**

Reviewer Comment 2.11: Results from different lithologies (starting at line 265) – to me, these observations suggest that lithology is a fundamental control on grain size dependency. If you're going to do multi-variate analysis to attribute grain size variability to each factor, can you tease apart the variability attributable to lithology first, and then discuss trends attributable to other factors? Ideally you'd do a multi-variate analysis for all 4 factors, but lithology isn't a continuous dataset, so this isn't really possible – teasing out the effects of lithology first would at least give you an idea of the relative importance of each factor including lithology.

Authors response: Based on the Kolmogorov-Smirnov test with a 5% significance interval, the distribution of grain size dependencies do not show significant differences for the different lithologies. It is not clear to us how we could test the variability of grain size dependencies by lithology alone. And, as the reviewer correctly stated, it is difficult to link rock types with numerical values that would allow proper statistical treatment. Although we agree with the reviewer that lithology likely plays an important role, it remains difficult to test quantitatively, as we discuss in the text. Our approach to classify into different rock categories that potentially behave differently is an attempt to make the influence of lithology visible. Therefore, we test the control and relative importance of slope, MAP and travel distance on each individual lithology separately to investigate which lithology may be more susceptible to certain processes. But because the data and trends are generally noisy, the introduction of subsets of the data reduces the number of observations and thus increases uncertainties.

**Revisions made:**

*Line 318: None of the lithologies revealed a significantly different grain size dependency distribution based on the KS-test.*

Reviewer Comment 2.12: Figure S5 suggests (or at least asserts – providing a calculation to back it up would be useful) that the altitudinal variation in 10Be production is not sufficient to explain the positive trend in 10Be with grain size. What about the other catchments? If coarse grains originated only at low elevations (i.e. were not transported from upper parts of the catchment), could that explain negative grain size dependency? Could this be a sediment transport story, rather than a landslide/depth shielding story? By ignoring the spatial variation in 10Be production, you're essentially assuming that sediment originates from all elevations – this assumption may not always be valid for all grain sizes, and it should probably be stated somewhere in the text.

Authors response: Good point. We did not report the catchments' total relief in our previous manuscript, which we included now. The total relief in the AZ, SG and NA catchments is low (<350 m), so that the variation in production rates is minor (<25%). We, therefore, think that

sediment provenance of different grain size of different elevations cannot explain the grain size trends of these catchments. We discussed the higher total relief and production rate differences for the La Campana catchment.

**Revisions made:**

Line 371: We propose that the existing or missing trends in the arid (AZ), semi-arid (SG) and temperate (NA) catchments are mainly related to differences in precipitation and the excavation depth of the erosion processes. These catchments show minor variations in mean basin slope, hence we do not expect big differences in erosion processes due to changes in slope alone. Furthermore, the limited relief of these catchments excludes differences in 10Be production rates and local sediment sources to influence observed differences in 10Be concentrations.

*Line 376: However, steeper hillslope angles and higher total relief may have overruled the effect of precipitation in the La Campana catchment.*

**2.3Technical comments:**

Reviewer Comment 2.13: Line 44 - modeling studies (Lukens et al., 2016) help constrain how big this bias could be (under- or overestimating erosion rates by a factor of 3 or more) – this might be worth mentioning here as context for the potential scope of the problem.

Authors response: Good point that clearly states the extent of the problem.

**Revisions made:**

Line 44: Where 10Be concentrations differ amongst grain size fractions, using a nonrepresentative grain size fraction could bias catchment-average denudation rates by a factor of 3 or more (Lukens et al., 2016).

Reviewer Comment 2.14: Line 107 - "Furthermore, fluvial processes can affect grain size fractions in a way that not all parts of a catchment are equally represented at a given sample location" – this is certainly true, but needs to be more clearly explained and certainly needs references. (You do this at the end of the paragraph, maybe this sentence just needs to move.)

Authors response: In order to clarify, we moved the sentences that describe how a non-representative sample might occur higher up.

**Revisions made:**

Line 134: Furthermore, fluvial processes can affect the grain size distribution at the sample location in a way that not all parts of a catchment are equally represented in different grain size fractions. For example, sediment provenance of grains from different elevations could play a role in catchments with heterogeneous rock types that produce different clast sizes or contain different quartz abundances (Bierman and Steig, 1996b; Carretier et al., 2015). An unequal representation of elevations in different grain size fractions may also result from hydrodynamic sorting, downstream abrasion and insufficient mixing of tributaries that drain different elevations (Carretier et al., 2009; Carretier and Regard, 2011; Lukens et al., 2016; Neilson et al., 2017). Combined with elevation-dependent 10Be production rates (and provided that denudation rates are constant), this could also result in grain size-dependent 10Be concentrations (Carretier et al., 2015; Lukens et al., 2016; Matmon et al., 2003). Reviewer Comment 2.15: Line 113 - "grain size" here refers to mineral grains, yes? Be careful/specific when discussing mineral grains vs clast sizes. These are different problems to consider, and arise for different reasons (lithologic controls vs. weathering/geomorphic process/etc.). I've run into a fair bit of confusion from readers/reviewers in my own work for just this reason, and the only advice I can give is that clarity and consistency of language around this distinction is paramount – I'd suggest changing "grain size" here to "mineral size".

Authors response: Good point, it may not have been always clear whether we're talking about mineral grain sizes or clast sizes; we improved this in the text. In this particular sentence we actually meant that rocks like schist may produce different clast sizes than granite, for example.

**Revisions made:**

Line 135: For example, sediment provenance of grains from different elevations could play a role in catchments with heterogeneous rock types that produce different clast sizes or contain different quartz abundances (Bierman and Steig, 1996b; Carretier et al., 2015).

Reviewer Comment 2.16: Line 118 - misplaced comma towards the end of the line (should go after "depth")

Authors response: Thanks. We removed this sentence from the revised manuscript.

Reviewer Comment 2.17: Line 123 - "process" should be plural

Authors response: Changed

**Revisions made:**

Line 147: Here, we consider mean basin slope as a topographic catchment attribute that controls denudation rates and the scouring depth of diffusive or deep-seated erosion processes.

Reviewer Comment 2.18: Line 246 - "Uncertainties in 10Be concentrations tend to be higher for samples from steeper hillslope angles (>10°), which is related to generally lower concentrations, i.e., higher denudation rates." This sentence is confusing. On my first read I thought you were suggesting that hillslope angle controlled 10Be uncertainties. Too many ideas in one sentence, break it up for the sake of clarity. Steeper hillslopes are eroding faster, which means they have lower 10Be concentrations. Uncertainties are larger for very low 10Be/9Be ratios.

Authors response: We understand the confusion. We restructured and clarified the sentence.

**Revisions made:**

Line 287: Uncertainties in 10Be concentrations tend to be larger for samples from steeper catchments (>10°), which may be related to higher denudation rates and therefore lower 10Be concentrations. Generally, uncertainties are larger for low 10Be/ $^{9}$ Be ratios.

Reviewer Comment 2.19: Line 248 - "nearly similar" is redundant, nix the "nearly"

Authors response: We prefer to be careful with stating that all samples have similar 10Be concentrations, in fact there is some minor variation. We replaced the word 'nearly' for 'relatively'.

**Revisions made:**

Line 290: 10Be concentrations are relatively similar across all grain size classes

Reviewer Comment 2.20: Line 277 - "Only in the arid. . ." Awkward sentence structure – consider flipping it around (Trends in 10Be only exist in the arid and Mediterranean catchments)

Authors response: That reads better indeed, we corrected that.

*Revisions made: Line 272: Only the arid (AZ) and Mediterranean (LC) catchments show a consistent, but noisy trend between* 10*Be concentrations and grain sizes.*

Reviewer comment 2.21: Line 287 - missing comma before figure reference

Authors response: Thanks, it was actually supposed to be a parenthesis.

Revisions made: Line 357: (Figure S5)

Reviewer Comment 2.22: Line 289 - Are the soil pit 10Be measurements also from Schaller et al. 2008? If so, cite them here. Referencing Figure 8 here would also be appropriate.

Authors response: Yes, all soil pit 10Be measurements are from Schaller et al., 2018. We included a reference to Schaller et al., 2018 and Figure 8 after each sentence discussing the comparison to the soil pits.

Reviewer Comment 2.23: Line 317 - "threshold slopes. . . where hillslopes cannot get any steeper" It's a nitpicky point, but I'd quibble with this language. Conceptually, yes, slopes steeper than the threshold "shouldn't" exist, but they certainly do at local scales. Your plots have slopes steeper than the threshold, even at basin-averaged scales, so your own data attest to the fact that hillslopes CAN get steeper.

Authors response: Good point; we deleted "at which hillslopes cannot get any steeper".

**Revisions made:**

Line 398: In these catchments, many hillslopes have likely reached the threshold hillslope angle of ~25-30°, at which denudation rates are dominated by the frequency of landslides (Larsen and Montgomery, 2012; Montgomery and Brandon, 2002; Ouimet et al., 2009).

Reviewer Comment 2.24: Line 356 - "We thus think" = awkward Authors response: Corrected

*Revisions made: Line 371: We propose, ....*

Reviewer Comment 2.25: Line 357 - missing "the" before "mixed soil layer"

Authors response: Corrected

*Revisions made: Line 382: ..the thickness of the mixed soil layer*

Reviewer Comment 2.26: Line 370 - don't start a sentence with "And"

Authors response: Corrected by combining both sentences.

**3. Comments on review of Anonymous Referee #3:**

This is an interesting and timely study presented by van Dongen et al., assessing the influence of the grain size used in detrital 10Be sampling and how it may bias measured concentrations and subsequently estimated catchment averaged denudation rates. The authors present new CRN data from a series of catchments in the Chilean Coastal Cordillera which span across a notable climatic gradient, to test the effect of precipitation on 10Be grain size dependence. They combine this with an analaysis of other metrics in these catchments (e.g. hillslope angle, lithology, abrasion) which are likely to produce grain size fractions with variable 10Be concentrations. Finally, a similar analysis is performed on a global dataset of 10Be samples and reported grain size fractions to test whether grain size dependencies exist. Overall, I found the paper interesting and relatively well written. I think there are some important messages concerning the conditions under which grain size dependency may bias 10Be sampling which can be drawn from the manuscript. With some clarification and moderate changes, I would support publication of the manuscript.

**3.1General comments**

Reviewer Comment 3.1: One of my first thoughts on reading this manuscript is the overlap with that of Binnie et al. (2007) Geology, although there is no reference to this study. One of the main finding from this earlier study was that denudation rate and slope gradient are broadly linear up to threshold hillslope gradients of  $\sim$  30 degrees. Beyond this, denudation rates are much more variable because of a transition from transport-limited to detachment limited denudation processes – i.e. steepening the hillslope beyond this has limited effect on catchment averaged denudation rate. In general, I felt there were a number of key references missing in the first few sections of this manuscript. I think it needs to be made clearer how the findings from this study are new/different from previous work.

Authors response: We included Binnie et al. (2007) in 3 text sections and aimed to improve links to previous work.

**Revisions made:**

Line 47: Some studies inferred that lower 10Be concentrations in coarse grains are caused by deep-seated erosion processes, such as landslides, which excavate material from greater depth where 10Be concentrations are lower (e.g. Aguilar et al., 2014; Belmont et al., 2007; Binnie et al., 2007; Brown et al., 1995; Puchol et al., 2014; Sosa Gonzalez et al., 2016a, 2016b; Tofelde et al., 2018; West et al., 2014).

Line 124: Because 10Be production rates decrease exponentially with depth (Gosse and Phillips, 2001), hillslope sediment that is excavated over a larger depth interval by landslides will obtain a larger variation in 10Be concentrations than sediment transported by diffusive processes near the surface (Aguilar et al., 2014; Belmont et al., 2007; Binnie et al., 2007; Brown et al., 1995; Puchol et al., 2014; Sosa Gonzalez et al., 2016a, 2016b; Tofelde et al., 2018; West et al., 2014).

Line 401: This conforms with previous studies that also found negative grain size dependencies which emerged from a transition of transport-limited to detachment-limited erosion processes and, therefore, deep-seated erosion processes (Binnie et al., 2007; Brown et al., 1995; Lukens et al., 2016; Reinhardt et al., 2007; Sosa Gonzalez et al., 2016a, 2016b; Tofelde et al., 2018).

Reviewer Comment 3.2: Production rates: In terms of CRN production rate varying measured concentrations – I'm not sure what the total relief across the Cordillera catchments are (could you possibly add this into Table 1?) or the CRN production rate that was used (please also add this in somewhere) for each catchment but it seems unlikely that this would have a significant impact unless there is some considerable relief?

Authors response: The total relief and differences in production rates are low for the AZ, SG and NA catchments, therefore we think this won't have a large effect. However, as discussed in the original manuscript, total relief and differences in production rates are much higher in the La Campana catchment, but these do not explain the positive grain size trend.

**Revisions made:**

We included total relief and the production rates used for the erosion rate calculation in Table 1 and 2.

Line 371: We propose that the existing or missing trends in the arid (AZ), semi-arid (SG) and temperate (NA) catchments are mainly related to differences in precipitation and the excavation depth of the erosion processes. These catchments show minor variations in mean basin slope, hence we do not expect big differences in erosion processes due to changes in slope alone. Furthermore, the limited relief of these catchments excludes differences in 10Be production rates and local sediment sources to influence observed differences in 10Be concentrations.

*Line 376: However, steeper hillslope angles and higher total relief may have overruled the effect of precipitation in the La Campana catchment.*

Reviewer Comment 3.3: Lithology: Just a quick query about quartz content of lithologies – I'm presuming most of these lithologies are fairly quartz rich such that this shouldn't bias any results (I'm thinking mostly about the mixed category)? Line 130 states some variations in mineralogy exist – do you have any maps/indication as to whether this is significant in terms of biasing quartz distribution across the catchment?

Authors response: These included mainly quartz-rich lithologies indeed. With the sentence "some variations in mineralogy exist" we meant variations between sites and not within sites. We, therefore, think no grain size dependencies arise because of mineralogy differences within a catchment. We clarified this in our manuscript.

**Revisions made:**

*Line 158: The catchments share a granodioritic lithology, though some minor variations in mineralogy exist between the sites (Oeser et al., 2018).*

Reviewer Comment 3.5: Catchment size: I'm not entirely convinced by the interpretation on L326. Looking at Figure 7b, there does appear to be some increase in negative grain size dependency in larger catchments but only really in sedimentary lithologies. Is the scale correct on the horizontal axis in Figure 7b (I thought only catchments <5000 km2 were considered)? In these larger catchments, I'd expect that these sedimentary rocks would abrade more quickly into finer grain size fractions, especially given these greater travel distances – perhaps these coarser and lower concentration grains are actually more locally produced (lower production rate if from lower elevations too possibly?). In the sections following Line 332, there is a counter argument that in really large catchments (the exact size would depend on lithology/abrasion thresholds I presume) the effects of grain size dependency are likely to

be less, as the majority of material should be abraded into sand? I think this comment rests upon whether the horizontal scale in Figure 7b is correct or not. It might be helpful to define what you consider as a large catchment here (>1,000 km2 etc.)?

Authors response: We indeed only included catchments of <5000km2, but we defined mean travel distance as the arithmetic mean distance that grains need to travel from all DEM grid cells in the catchment to the sample location. This metric is strongly correlated with catchment area (see Figure S6). Especially with the new phi-based grain size classes, the effect of mean travel distance is only visible in sedimentary catchments. We fully agree that abrasion may induce elevation-dependent grain size fractions, especially in catchments with contrasting 10Be production rates. This is what we wrote in the original text, but improved the text to make this point clearer. It should be noted however that quite often these different controls combine. Large differences in production rates result from large values in catchment relief, which typically comes along with steep slopes. That's why it is so difficult to disentangle the different controls.

**Revisions made:**

Line 410: Our results revealed a weak negative control of sediment travel distance on grain size dependencies, however no significant relationships were found. The negative control is strongest for sedimentary catchments in which negative grain size dependencies appear to be more frequent in catchments with long sediment travel distances (Figure 8b). For sedimentary catchments the most negative grain size dependencies appear when travel distances exceeded the abrasion threshold. Possibly the lower rock strength of sedimentary rocks promotes the breakdown into smaller particles and increases the grain's sensitivity to abrasion (Attal and Lavé, 2009; Sklar and Dietrich, 2001).

Line 417: As travel distance scales with elevation (Figure S6) and, therefore, 10Be production rates, sediment from high elevations may have inherently higher nuclide concentrations (Lal, 1991). In contrast, coarse grains which experienced less abrasion may origin from lower elevations, with lower 10Be production rates. This elevation-dependence of certain grain size fractions may induce a negative grain size-dependency.

Reviewer Comment 3.6: Lupker et al. (2012) considered multiple grain size fractions in samples taken from a number of Himalayan catchments at the mountain front and found no systematic trend or differences in 10Be concentrations as a function of sand grain size (which makes up the majority of the sediment load). With increasing catchment area, one would expect the concentrations measured in the sand fraction to be more representative of the total catchment. As catchments get larger, there are also likely to be different erosional processes operating within in which may influence 10Be concentrations (see Dingle et al., 2018). For example glacial shielding (which will offset any difference in production rate as a function of elevation), glacial erosion, sediment recycling and 'hotspots' of erosion which may be driven by spatial variations in climate which can occur across sufficiently large catchments (e.g. localised storms), or parts of the catchment which undergo higher rates of rock uplift and are more susceptible to landsliding. There are then also issues relating to temporary storage (even just within the channel itself, or within large landslides) within increasingly larger catchments. I think you touch upon this in Line 59.

Authors response: These are all good points and we fully agree. Teasing apart all of these potential controls is the main difficulty we encountered in our study of the global compilation. To effectively reduce some potential influences, we decided to include only catchments with a limited size (<5000 km2), but not too small so that we would have no data to analyse.

Reviewer Comment 3.7: Temporal effects: One of the key aspects I feel this manuscript currently overlooks is a discussion on how representative the Cordillera samples are. These catchments are small (<10km2) and experience landsliding – how likely is it that these samples are influenced by the stochastic nature of sediment delivery from these landslides (e.g. Niemi et al., 2005 – I noticed that this paper wasn't referenced anywhere). Do you have truly 'representative' samples and how stable are these concentrations in different grain sizes in time? Is material generated by these deposits likely to be well mixed into the suspended/bed material load (especially given such short transport distances), or is it likely to overwhelm the catchment-averaged signal depending on factors such as the time since the last event/time since significant mobilisation of landslide material.

Authors response: To give a complete documentation of processes that may control the 10Be concentrations measured in river sediment, we additionally discussed a spatially and temporally representative sample. We think that, of the 4 coastal cordillera catchments, potentially the La Campana catchment may be influenced by landsliding, and we included this in the text.

**Revisions made:**

*Line 110:* If mixing within the channel is incomplete, single tributaries or local inputs of sediment (e.g. landslides) may dominate the grain size distribution (Binnie et al., 2006; Neilson et al., 2017; Niemi et al., 2005; Yanites et al., 2009).

*Line 121: A certain sample location provides a spatially and temporally representative sediment sample when the sediment from different sources is sufficiently mixed (Binnie et al., 2006; Neilson et al., 2017; Niemi et al., 2005; Yanites et al., 2009).*

*Line 353: Deep-seated erosion processes and insufficient mixing in a small-sized catchment may make a sample non-representative for the entire catchment (Niemi et al., 2005; Yanites et al., 2009).*

Reviewer Comment 3.8: Grain size: It took me a while to get my head around what the normalized grain size statistics actually represent, especially given the ranges may have varied between the studies looked at (e.g. where only >2 mm was stated, values were forced into 2-4 mm). If I have this right, the grain size fractions presented in each study could influence the normalized grain size you calculate if these fractions were inconsistent between studies? It would be really nice to see metrics plotted against absolute grain size (maybe somewhere in the SI) given you have this information available.

Authors response: See our response on Reviewer comment 2.10.

**Revisions made: See revisions at Reviewer comment 2.10.**

Reviewer Comment 3.9: Another point which I think Reviewer 2 has also commented on – what is a representative grain size of a catchment (thinking about Line 28-29 in the abstract in particular), and is this what is being sampled on the river bed (e.g. Figure S2 shows that some of the catchments capture more of the CDF than others)? Interesting that your bedload GS in Chile is much more bimodal than in either Italy or the SGM datasets (Figure 1).

Authors response: A good question, no doubt. In fact, we chose this to introduce the topic (Figure 1). Yet, we don't pretend to know (neither before or after this study), what a

representative grain size fraction is, as this requires knowledge of the grain size distribution of all erosion sources within a catchment and what their relative contribution to the grain size distribution of the entire catchment is. Nevertheless, if there is no grain size trend with 10Be concentrations, then measuring the concentration in any grain size would provide an unbiased estimate. Yes, the bimodality is striking. And it is likely related to a large amount of catchments that have been measured along the climate gradient, which included catchment which had a predominantly fine grain size distribution and catchments which had a coarser one. However, after reconsidering this figure, we decided to delete the Chile data, because it is unpublished data.

**3.2Specific comments:**

Reviewer Comment 3.10: Line 52 – Which other studies? Add some references

Authors response: We moved the references that were cited after the following sentence forward for clarification.

**Revisions made:**

Line 54: However, other studies found that grain size reduction by abrasion during fluvial transport, or spatial variations in the provenance of different grains sizes can additionally account for grain size-dependent 10Be concentrations (*Carretier et al., 2009; Carretier and Regard, 2011; Lukens et al., 2016; Lupker et al., 2017; Matmon et al., 2003).*

Reviewer Comment 3.11: Line 87-94 - I found this paragraph a little wordy. "when the transport capacity of the water is too low" - too low for what? Please be more specific.

Authors response: We revised this paragraph.

**Revisions made:**

Line 107: Once the sediment has reached the channel, processes like downstream abrasion, selective transport and mixing of sediment sources control the grain size distribution at the sample location. If mixing within the channel is incomplete, single tributaries or local inputs of sediment (e.g. landslides) may dominate the grain size distribution (Binnie et al., 2006; Neilson et al., 2017; Niemi et al., 2005; Yanites et al., 2009). Downstream abrasion and selective transport result in a progressively smaller grain size distribution. Abrasion wears off the outer layers of clasts (Kodoma, 1994; Sklar et al., 2006) and depends on the travel distance and velocity as well as the lithology of the clasts (Attal and Lavé, 2009). Selective transport preferentially deposits coarse grains when the transport capacity of water is low (Ferguson et al., 1996; Hoey and Ferguson, 1994) and thus further changes the grain size distribution.

Reviewer Comment 3.12: Line 95-97 - "Any process that transports different grain sizes, from areas in a catchment with contrasting. . ..". You could also say the same for grains of the same size from different parts of the catchment. I feel that this paragraph could do with a little more work generally. For example, in Line 107 you discuss variations in 10Be concentrations in soil as a function of whether the landscape is eroding quickly or not. There is the argument that in more rapidly eroding landscape you would only expect larger variations in concentrations (due to removal of material from depths greater than the attenuation length) if a concentration profile is fully developed. In rapidly eroding landscapes, you may just end up with upper layers characterized by relatively (uniformly) low concentration material?

The jump to the metrics you propose to look at in L117 onward feels quite big. It would be nice to see a clearer build up to this in the paragraph beforehand (Line 70 onwards) so that it is obvious why these metrics have been chosen.

Authors response: We agree and made this paragraph more general to emphasize that these variations in 10Be concentrations may emerge in sediment in general. In order to make a clearer link to the metric, we included a new paragraph which describes why we chose the metrics. We are not fully sure we understand the comment on line 107. But in rapidly eroding landscapes soils and thus the mixed layer also tend to be thinner, so the likelihood of excavating material from greater depth with a lower concentration that the surface material exists. Whether the absolute concentrations are high or low is less important, but the relative difference matters.

**Revisions made:**

Line 124: Because 10Be production rates decrease exponentially with depth (Gosse and Phillips, 2001), hillslope sediment that is excavated over a larger depth interval by landslides will obtain a larger variation in 10Be concentrations than sediment transported by diffusive processes near the surface (Aguilar et al., 2014; Belmont et al., 2007; Binnie et al., 2007; Brown et al., 1995; Puchol et al., 2014; Sosa Gonzalez et al., 2016a, 2016b; Tofelde et al., 2018; West et al., 2014).

Line 131: Eroded sediment from these layers is thus expected to have uniform 10Be concentrations. In rapidly eroding and arid landscapes, however, soils are typically very thin or absent, and the eroded sediment likely yield larger variations in 10Be concentrations (Figure 2).

**Line 144:* **2.3** *Catchment attributes that potentially control grain size-dependent* 10*Be concentrations**

Based on the above review of processes that may influence grain size-dependent 10Be concentrations, we identified the catchment attributes mean basin slope, mean travel distance, mean annual precipitation (MAP), and lithology that we will focus on in our study. Here, we consider mean basin slope as a topographic catchment attribute that controls denudation rates and the scouring depth of diffusive or deep-seated erosion processes. We selected MAP because of its effect on both weathering rates and the scouring depth of erosion processes and lithology because it affects chemical weathering rates, the grain size of individual minerals and the susceptibility to hillslope failure. Finally, we selected mean travel distance of sediment as a metric for fluvial processes that are transport-dependent (e.g. abrasion and hydrodynamic sorting).

Reviewer Comment 3.13: Line 174 – what happens if you remove these studies (stated only as >2 mm) from your statistics? I appreciate this may remove a large number of points but might be interesting to see.

Authors response: Yes, it would remove a large number of catchments and it doesn't change the overall pattern.

Reviewer Comment 3.14: Line 246 – Are these uncertainties relating to error/uncertainty in the laboratory measurements or variability in measured concentrations? Either way, uncertainty and variability are different so please clarify!

Authors response: The error bars on each individual sample are the analytical uncertainties, the variability discussed in the text represents the variability of 10Be concentrations of all grain sizes within a catchment. We made this distinction clearer in the text and figure captions.

**Revisions made:**

Line 274: The  $2\sigma$ -variability of 10Be concentrations measured in all grain size fractions deviates  $\pm 18\%$  from the mean (Figure 5). In the Mediterranean catchment (LC), the 10Be concentrations of all grain size fractions vary up to  $\pm 40\%$  from the mean and display a noisy but positive grain size dependency, i.e., increasing 10Be concentrations with increasing grain size (Figure 5). In both the semi-arid (SG) and temperate catchments (NA), the  $2\sigma$ -variability in 10Be concentrations is low ( $\pm 12\%$  and  $\pm 14\%$ , respectively) and rather unsystematic (Figure 5).

Reviewer Comment 3.15: Line 250 – While no pattern in MAP, I wonder whether the frequency of large storms is a factor that is likely to be important? Is it fair to assume that all sediment generated by landsliding in these catchments is immediately evacuated from the catchment and there is no preferential mobility of coarser/finer material (i.e. it might take a large storm to mobilise the coarsest material which may only happen a few days of the year?).

Authors response: Likely yes, this is a good consideration for future work. However, most of these catchments do not contain any weather stations. Even if global datasets of storm frequency or related metrics exist, it is not clear to us how well they depict actual conditions. In the more arid regions, large storms may not be included in historic records and thus be biased. One solution could be to analyse a combination of records (long), satellite observations (short), and climate model results. However, this is outside the scope of the current study.

Reviewer Comment 3.16: Line 254 – Is 54.8% really 'significant'? In general I found some of the statistics a little weak and definitions of coarse/fine not fully stated. For example, on line 234 and Figure 4, you state that only the AZ and LC catchment show consistent trends between 10Be and grain size. When I look at Figure 4 I see a lot of scatter/variability!

Authors response: The 54.8% refer to the percentage of catchments that have error bars in grain size trends that do no overlap with the zero line. This number compares to 32.8% of the data that lies above the line and 11.0% that is within error of no grain size trend. We clarified this in the text. With consistent trend we meant that one can state that there is an overall positive or negative tendency, however this trend can be noisy. We also improved this in the text.

**Revisions made:**

*Line 297: Overall, we observe more sample sets that display significantly (i.e. error bar does not overlap with 0) negative (56.2%) trends in grain size-dependent 10Be concentrations, than positive (32.8%; Figure 8).*

*Line 272: Only the arid (AZ) and Mediterranean (LC) catchments show a consistent, but noisy trend between* 10*Be concentrations and grain sizes.*

Reviewer Comment 3.17: Line 264 - 'Partly accentuated and partly muted' - this is a very confusing sentence! Authors response: We removed the sentence. Reviewer Comment 3.18: Line 278 – 'In both catchments the 10Be concentrations of river sediment correspond to concentrations measured deeper within soil profiles' – looking at AZ and LC in Figure 8 it looks like the river concentrations correspond to the concentrations measured in the upper 1m of the soil profile (AZ), not material from greater depths. In LC, it looks like all of the river grain size fractions are consistent with concentrations measured below 1m in the soil puts, suggesting no grain size dependence. Instead this seems to suggest that all of the sampled sediment is overwhelmed by material excavated from depth? Could the fact that the river sediment concentrations are lower than those in the soil pit (line 288) simply reflect the fact that the concentrations measured in the soil pits are not representative of the entire catchment?

Authors response: Agreed, greater depth in the arid (AZ) catchment only means ~1 m depth, we clarified this in the revised manuscript. We also discussed the possible offset of 10Be concentrations in the channel sediment of La Campana. And finally, yes, we agree that the soil pits may not be representative for the entire catchment. We added a sentence that admits this possibility.

**Revisions made:**

Line 345: In the arid catchment (AZ), both the negative grain size dependencies and the fact that 10Be concentrations correspond to concentrations at ~1 m depth in the soil profiles suggest that erosion processes (e.g. rock falls, landslides, gully head retreat), which excavate sediment from intermediate to greater depth during rare precipitation events or earthquakes (e.g. Mather et al., 2014; Pinto et al., 2008), may occur in this catchment.

Line 351: This suggests that the catchment experiences faster erosion processes compared to the location of the soil pit, which is confirmed by debris flow scars observed at high elevation in the catchment (Figure S5).

Reviewer Comment 3.19: Line 300 - I find this sentence undermines the study slightly...maybe consider rephrasing 'our new samples from the Chilean Coastal Cordillera demonstrate minor variations in 10Be concentrations'.

Authors response: Agreed, we removed this first sentence in our discussion section.

Reviewer Comment 3.20: Table 2 – what is the superscript b referring to in the last column title?

**Authors response: Deleted**

Reviewer Comment 3.21: Figure 2 – Line 707 – 'constant' or maybe uniform?

Authors response: Changed

**Revisions made:**

*Line* 862: *Bioturbation in landscapes with thick soil-mantles results in a well-mixed soil layer with a uniform10Be concentration, which, in isotopic steady state, is equal to the surface concentration.*

Reviewer Comment 3.22 Figure 6 - I suspect this is one of the key figures for the paper but find it difficult to follow. There is a lot of information in there.

Authors response: See our response at Reviewer Comment 1.6.

Reviewer Comment 3.23: Figure 9 – 'results given for all lithologies combine' – should be 'combined' Authors response: We changed the figure caption.

*Revisions made: Line 914: Coloured symbols depict lithological classes.*

**4. Revised manuscript with track changes**

[revised manuscript text omitted]

To compare data from different study areas with different 10Be production rates and from studies that measured
 different grain size classes, we normalized the 10Be concentrations (±2σ analytical uncertainties) and grain size classes by the arithmetic mean concentration and grain size class of all samples from the same catchment.

- To assess the influence of the identified catchment attributes Based on our discussion review of processes that potentially induce influence the grain size distribution and 10Be concentrations of grain size dependent 40Be concentrations in river sediment (paragraph 2), we focus on the following catchment attributes:study the effect
- 250 of-mean basin slope, mean travel distance, mean annual precipitation, and lithology (Section 2.3) on grain-size trends in the global compilation, -in our further analysis. Www used a 90-m resolution SRTM DEM (Jarvis et al., 2008), to-We obtained upstream areas based on the published sample coordinates and using the flow routing tools of the TopoToolbox v2 (Schwanghart and Scherler, 2014). We recalculated the published sample

characteristicstopographic parameters: catchment area, mean basin slope, total relief (maximum elevation -

255 minimum elevation) and the mean travel distance of grains-sediment to the sample location, which is calculated as the-(\_arithmetic mean travel distance of all pixels in the catchment to the of travel distances calculated between each pixel in the catchment and the sampling pointe location). The agreement between the published and recalculated values-topographic parameters is good, and minor deviations likely result from differences in DEM resolution (Figure S3). We obtained an estimate of mean annual precipitation (MAP) in each catchment using the 0.25°-resolution gridded precipitation data set from the Global Precipitation Climatology Centre (Meyer-Christoffer et al., 2015). To classifydetermine catchment lithology we used the Global Lithological Map

(GLiM; Hartmann & Moosdorf, 2012) in combination together with the lithology reported in the original

publications. We defined four different lithological classes: sedimentary, magmatic, metamorphic and mixed (>3 different lithologies rock types in a catchment).

265 Next, wWe used Sternberg's Law to estimate the extent of abrasion of bedload sediment during fluvial transport, to define a travel distance threshold after which abrasion becomes significant:

$$D(L) = D_0 e^{-\alpha L}$$
(1)

Using equation 1, we calculated the grain size D at the sample location, which remains is derived from an initial grain size  $D_0$  at the source, that travelled a distance L and decreased in size according to at a rate given by the

- 270 reduction coefficient  $\alpha$  (Kodama, 1994; Kodoma, 1994; Lewin and Brewer, 2002; Sklar et al., 2006; Sklar and Dietrich, 2008). The reduction coefficient depends on both grain velocity and lithology. Rocks with low tensile strength decrease-reduce faster in size during transport compared tothan rocks with high tensile strength (Attal and Lavé, 2009). We chose the reduction coefficients based on literature values for different-field settings (sedimentary rocks:  $\alpha = 0.0003$  m-1, magmatic rocks:  $\alpha = 0.0002$  m-1, metamorphic rocks:  $\alpha = 0.0001$  m-1),
- 275 which are typically higher than experimental studies due to different particle collision dynamics and the lack of weathering in experimental studies (Sklar et al., 2006). We defined considered the effect of abrasion to be negligible when a grain size at the erosion sourceat the sample location ( $D_0$ ) still falls in the same Wentworth phi-grain size class as at its erosion sourceat the sample location ( $D_0$ D). E.g., fFor abrasion to be significant, a grain size of 2 mm at the erosion source, for example, must be reduced by more than 50%, hence be smaller
- 280 than 1 mm at the sample location to fall in a lower phi-grain size class. This results in abrasion thresholds for sedimentary, magmatic, and metamorphic rocks of 2300 m, 3500 m, and 7000 m, respectively. For catchments underlain by mixed lithologies, the abrasion threshold lies between 2300 m and 7000 m (Attal and Lavé, 2009). We quantified the relationship between grain size and 10Be concentrations by calculating a 'grain size'.
- dependency' for each sample set (Figure 4). This, which\_corresponds-is the slope of a linear fit through the 10Be
   concentrations of different grain size classes. to the regression coefficient of a linear model fitted to all samples
   within a sample set. A positive grain size dependency indicates higher 10Be concentrations in coarser grains, and
   <del>vice versa.</del> To account for uncertainties in 10Be concentrations and for grain size ranges, we used a Monte Carlo
- approach (n=10,000) to randomly select a point between the mean  $\pm \frac{2\sigma}{2\sigma}$ -analytical error 10Be concentrations and the analysed grain size range. We thus obtained a mean  $\pm \frac{2\sigma}{2\sigma}$ -standard deviation grain size dependency for each catchment. A positive grain size dependency indicates higher 10Be concentrations in coarser grains, and vice
- versa. Next, we used the Kolmogorov-Smirnov Test (KS-test, 5% significance interval) to test whether particular mean

basin slope, MAP or sediment travel distance classes showed a significantly different distribution of grain size dependencies (Kolmogorov, 1933; Smirnov, 1939). NextFinally, we calculated linear regression the statistics of

- 295 a linear regression between the grain size dependency values and the catchment attributes mean basin one or more of the slope, MAP and mean travel distance catchment attribute and applied a multivariate linear regression model including the effect of all 3 catchment attributes.—We did this for the entire dataset and for each individual lithology. Finally, weAs part of the multivariate statistics, we calculated the relative importance (RI) of mean basin slope, mean travel distance and MAPall catchment attributes, using the LMG approach
- 300 (Lindeman et al., 1980), of the 'Relaimpo' R studio-package (Grömping, 2006), which This provides is the percentage of contribution of each catchment attribute to the multivariate regression model R2. We used the LMG approach (Lindeman, Merenda and Gold, 1980), which is part of the 'Relaimpo' R studio package.

**4.5. Results**

**4.15.1 Chilean Coastal Cordillera**

- 305 The measured 10Be concentrations in the most arid catchment (AZ) range from 2.8 to  $4.6 \times 10^5$  atoms (g quartz)- 1, resulting in catchment-average denudation rates of  $5.8 \pm 0.7$  to  $10.1 \pm 1.1$  mm kyr-1 (Table 2<del>, Figure 4</del>). In the semi-arid catchment (SG), the 10Be concentrations range from 3.6 to 5.2  $\times 10^5$  atoms (g quartz)-1, which corresponds to catchment-average denudation rates of  $7.5 \pm 0.8$  to  $11.0 \pm 1.4$  mm kyr-1 (Table 2<del>, Figure 4</del>). The 10Be concentrations in the Mediterranean catchment (LC) are a factor 10 lower compared to the other 310 catchments and range from 0.2 to 0.6  $\times 10^5$  atoms (g quartz)-1, which results in catchment-average denudation rates of  $103.7 \pm 12.4$  to  $384.1 \pm 54.5$  mm kyr-1 (Table 2, Figure 4). The temperate catchment (NA) yielded 10Be concentrations ranging from 1.8 to  $2.9 \times 10^5$  atoms (g quartz)41, resulting in catchment-average denudation rates of  $24.0 \pm 2.6$  to  $40.2 \pm 4.5$  mm kyr-1 (Table 2<del>, Figure 4</del>). Only the arid (AZ) and Mediterranean (LC) catchments show a consistent, but noisy trend between 10Be concentrations and grain sizes. In the arid catchment (AZ), 10Be 315 concentrations are decreasing with increasing grain size, and tT-he 2 $\sigma$ -variability of  $the 10^{-10}$ Be concentrations measured in all grain size fractions deviates  $\pm 18\%$  from the mean (Figure 5), with a  $2\sigma$  deviation of -18% from the mean. In the Mediterranean catchment (LC), the 10Be concentrations of all grain size fractions deviate-vary up to  $\pm -40\%$  from the mean and display a noisy but positive grain size dependency, i.e., they increasing 10Be concentrationse with increasing grain size (Figure 5). In both the semi-arid (SG) and temperate catchments 320 (NA), the  $2\sigma$ -deviations between  $2\sigma$ -variability in 10Be concentrations is are low ( $\pm$ -12% and  $\pm$ -14%, respectively) and rather unsystematic (Figure 5). The smallest grain size fractions (0.5-4 mm) in the semi-arid
- catchment (SG) show a decreasing trend, but this trend increases again for coarser grain size fractions (4-32 mm). In the temperate catchment (NA), the-10Be concentrations are uniform in the five smallest grain size fractions (0.5-16 mm), but this trend breaks down at the two largest grain size fractions (16-64 mm), which have
   lower 10Be concentrations.

**4.25.2 Global compilation**

The global compilation includes 62-73 catchments covering a wide range of different hillslope angles, sediment travel distances, -and-MAP and lithologies (Figure 6). Figure 7 shows the data of all catchments, classified in 4 slope classes and colour-coded by lithology. Each box represents the normalized 10Be concentrations ± analytical uncertainties and the grain sizes range of a single sample. -of all catchments, for different classes of mean hillslope angle, and colour coded by mean annual precipitation. Uncertainties in 10Be concentrations tend to be higher larger for samples from steeper hillslope anglescatchments (>10°), which may be related to higher denudation rates and therefore lower 10Be concentrations, i.e., higher denudation rates. In catchments with mean basin hillslope angles <10°, 10Be concentrations are nearly relatively similar across all grain size classes, -. In steeper hillslope classes, coarse grains reveal lower 10Be concentrations compared to fine grains, with the largest deviations in coarse compared to fine grains. (Figure 7). We discern no pattern related to mean annual precipitation\_lithology from this figure but we emphasize that rmagmatic catchments are more abundant in

340 shallow sloping catchments, whereas metamorphic catchments are more abundant in steep catchments...

Figure 8 (and Figure S4, for plots separated by lithology) shows the grain size dependencies for allof individual catchments, resulting from the regression coefficientslope of a fitted linear modellinear fit to the normalized data from each catchment10Be concentrations of all grain size classes (describedsee methods section and in Figure 4). Overall, we observe more sample sets that display significantly (i.e. error bar does not overlap with 0) negative

- 345 (54.856.2%) trends in grain size-dependent 10Be concentrations, than positive (25.832.8%; Figure 8). 19.411.0% of the sample sets have grain size dependencies that are not significantly different from zero, and i.e., thus reveal no trendgrain size dependency. Furthermore, positive negative grain size dependencies trends are typically weaker stronger (i.e. lower higher absolute grain size dependencies differences between grain sizes) than negative positive trendsgrain size dependencies.
- The calculated grain size dependencies of individual catchments reveal a significant breakpoint at a mean hillslope angle of ~15° (KS-test, (Figure 8a). Catchments with mean hillslope angles <15° reveal a distribution with predominantly weak grain size dependencies. Steep catchments with hillslope angles >15° show a wider distribution with predominantly negative grain size dependencies (62.3% significantly negative). 70.0% of the catchments that exceed the threshold hillslope (>25°)The scatter and amount of negative grain size dependencies
- 355 (i.e., lower 10Be concentrations in coarse grains) increase at mean hillslope angles >15°, with the most negative grain size dependencies in catchments of >25°. have significantly negative grain size dependencies. Our analysis of sediment travel distance shows that the amount and magnitude of negative grain size dependencies slightly predominantly occurincrease at longer sediment travel distances (Figure 8b). However, catchments that exceeded the abrasion threshold (sedimentary: 2300 m, magmatic: 3500 m, metamorphic: 7000 m, mixed: 2300-
- 360 7000 m) show no significantly different grain size dependency distribution based on the KS-test. Finally, the data suggests a slightly increasing amount and magnitude of negative grain size dependencies with increasing MAP. Humid catchments (MAP >2000 mm yr-1) reveal a distribution of predominantly (90%) significantly negative grain size dependencies, which is significantly different (KS-test) from catchments with MAP <2000 mm yr-1 the highest number of negative grain size dependencies are found for very dry (<100 mm yr-1) and very
- 365 wet catchments (>2000 mm yr-1) (Figure 8c). However only a low number of catchments with MAP >2000 mm yr-1 compose the distribution.–When differentiating the catchments by lithology, the influences of hillslope angle, travel distance, and MAP are partly accentuated and partly muted. Catchments underlain by sedimentary and metamorphic rocks -mixed lithologies (75.0%) show the largest numbermost of significantly negative grain size dependencies (66.7% and 65.4%, respectively), followed by catchments underlain by mixed lithologies
- 370 (50.0%)-sedimentary and metamorphic rocks (both 60.0%). The number of significantly negative grain size dependencies is lowest for catchments underlain by magmatic lithologies (26.737.5%). None of the lithologies revealed a significantly different grain size dependency distribution based on the KS-test.

Linear regressions of grain size dependencies as a function of mean basin slope revealed significantly negative trends for all lithologies combined (p = 0.002) and for metamorphic catchments (p= 0.017) but not for the other
 lithologies alone (Table S4, Figure S4). MAP showed a significantly negative relationship with grain size dependencies for all lithologies combined (p= 0.007), and for catchments underlain by magmatic (p= 0.006) lithologyies. This trend, hHowever, the trend for magmatic catchments mainly results from one two-negative data points at higher MAP. No significant linear trends emerged between mean travel distance and grain size dependencies offor any of the lithologies.

When considering the combined influence of the-mean basin slope, MAP and mean travel distanceall studied factors (i.e., slope, travel distance and MAP) with a multivariate linear model, we found that the variance of all lithologies combined is significantly described (p=0.004) by two out the combination of theall 3 factors, but that the , however explained variance iwas low\_( $R^2=0.1906$ , Table S5). Most of the variance is significantly

- 385 explained-related by-to mean basin slope (relative importance, RI = 9.1%), followed by MAP (RI = 7.6%), whereas mean travel distance revealed no significant contribution (Table S5, Figure 9). Additionally, the mFurthermore, multivariate models yielded significant results when considering only for-magmatic (p= 0.031, R2= 0.552) and metamorphic catchments (p= 0.077, R2= 0.276)-showed\_significant results. In magmatic catchments, aA large proportion of the variability in magmatic catchments is significantly deascribed by-to
- $\frac{MAP (RI = 51.4\%), \text{ again-which resulting s from a single-the above mentioned negative data point at higher MAP. Finally, the largest proportionmost of the variability in magmatic catchments was significantly described by mean basin slope (RI = 22.0\%). Multivariate statistics yielded insignificant results for mixed and sedimentary lithologies, possibly, due to too few catchments to disclose unambiguous trends.$
- most of the variance of catchments with mixed lithology is explained by travel distance (RI = 25.0% ± 4.7%)
   and MAP (RI = 17.5% ± 4.7%) (Table S5, Figure 9). In magmatic catchments, a large proportion of the variance is explained by MAP (RI= 43.8% ± 13.1%). This trend, however, mainly results from two negative data points at higher MAP. The RI statistics yielded low values for the other lithologies, possibly, due to too few catchments to disclose unambiguous trends.

**5.6. Discussion**

**400 **5.16.1** Grain size-dependent 10Be concentrations in the Chilean Coastal Cordillera**

The sampled catchments on a-the climatic gradient in the Chilean Coastal Cordillera only show a only minor variations in 10Be concentrations, regardless of grain size. Systematic trend of 10Be concentrations with grain size in the arid (AZ) and Mediterranean catchments (LC). average 10Be production rate is therefore -35% higher than the production rate at the 10Be depth profile. In both catchments, the 10Be concentrations of river sediment 405 correspond to concentrations measured deeper withinin the subsurface of the soil profiles (Figure 10; Schaller et al., 2018). Because the difference between 10Be production rates of the catchment on average and at the soil profiles is small (<10%), we can compare measured 10Be concentrations directly. In the arid catchment (AZ), both the negative grain size dependencies and the fact that 10Be concentrations correspond to concentrations deeper-at ~1 m depth in the soil profiles suggest that deep seated erosion processes (e.g. landslides, debris 410 flowsrock falls, landslides, gully head retreat), which excavate sediment from intermediate to greater depth during rare precipitation events\_or earthquakes (e.g. Mather et al., 2014; Pinto et al., 2008), may occur in this catchment. All of the measured river sediment-10Be concentrations in river sediment in the Mediterranean catchment (LC) are considerably lower than the surface concentrations those measured in the mixed soil layer of two measured at the surface in the of soil profiles within their close proximity of catchment (Figure 10; 415 Schaller et al., 2018), This-hi suggests that the catchment experiences faster nting on fast eerosion processes compared to the location of the soil pit, whichich is confirmed by high elevation debris flow scars observed at high elevation in the catchment (Figure S5). Deep-seated erosion processes and insufficient mixing in a smallsized catchment may make a sample non-representative for the entire catchment (Niemi et al., 2005; Yanites et

- (LC) contradicts with observed evidence of debris flows in this catchmentthis hypothesis (Figure S5), as these debris flows would presumably also eexcavate material coarse grains from greater depth. Furthermore, hHigher 10Be production rates at the elevation where debris flows originate, and the condition that coarse grains only origin from that area cannot account for the positive grain size dependency alone (Figure S5). It is also notable that all of the measured river sediment concentrations are considerably lower than those measured in the mixed
- 425 soil layer of two soil profiles within the catchment. Without being able to clarify this issue, the lower 10Be concentrations of river sediment, combined with the observed greater scatter in the positive grain size dependency may hint at selective transport and longer residence times of coarse grains at higher elevations. In contrast, tThe 10Be concentrations in river sediments from the semi-arid (SG) and temperate (NA) catchments
- 430 show little variations and are similar to concentrations measured near the surface in soil pits (Figure 10; Schaller 430 et al., 2018)(Schaller et al., 2018). Within the temperate catchment (NA), the uniform 10Be concentrations in grains <16 mm, suggests that these originate from the ~70 cm thick mixed soil layer, whereas the lower 10Be concentrations in grains >16 mm suggests these may be derived from below the mixed layer (Figure 10; Schaller et al., 2018)(Figure 8). In the semi-arid (SG) catchment, the measured samples from the channel show
- higher 10Be concentrations compared to the mixed layer of the south-facing hillslope (Figure 10; Schaller et al., 2018). This suggests that grains are unlikely to be derived from greater depth, where 10Be concentrations are lower.

We think we can attribute propose that the existing or lackingmissing trends in the arid (AZ), semi-arid (SG) and temperate (NA) catchments are mainly related to differences in precipitation and the excavation depth of the

similar 10Be concentrations compared to those measured in the mixed soil layer of the north-facing hillslope and

- 440 erosion processes. These catchments show minor variations in mean basin slope-and total relief, but-hence we do not expect big shifts-differences in erosion processes due to changes in slope alone. Furthermore, the limited relief of these catchments excludes differences in or a control of contrasting-10Be production rates and local sediment sources to influence observed differences in 10Be concentrations. Also, the difference with-10Be production rates at the soil profiles was small (<10%), which allows us to compare catchment average-10Be
- 445 concentrations to 10Be concentrations in the depth profiles to infer the depth were sediment was excavated from. (Schaller et al., 2018). However, steeper hillslope angles and -a-higher total relief may have overruled the effect of precipitation in the La Campana catchment. We do not expect a control related to the different catchment sizes in any of the catchments, because granitic rock have a low abrasion breakdown rate (Attal and Lavé, 2009) and the mean travel distances were small (<1 km).
- In summary, we think our new samples from the Chilean Coastal Cordillera only demonstrate minor variations in 10Be concentrations. The reflect the suggest an influence of MAP on grain size-dependent 10Be concentrations only -in the most-arid and most-humid catchments may-by its effect on reflect the thickness of the mixed soil layer and the scouring depth of erosion processes that transport larger grains from below the mixed soil layer.z

**5.26.2 Grain size-dependent 10Be concentrations in the global compilation**

**455 5.2.16.2.1 Mean basin slope**

The effect of mean basin slope on grain size-dependent 10Be concentrations is apparent as weak grain size dependencies in gently sloping catchments, and predominantly negative grain size trends in steep catchments

(Figure 7, Figure 8a). Mean basin slope may also-control grain size-dependent 10Be concentrations through its effect on the thickness of soils and the scouring depth of erosion processes.

[revised manuscript text omitted]

- 10Be production rates., Tthis elevation-dependence of certain grain size fractions may induce a negative grain size-dependency, Secondly, Hif abrasion were to reduce river sediment of decimetre- or meter-scale to sand size, the shielded interiorcentre of such clasts would have lower concentrations (Carretier and Regard, 2011; Lupker et al., 2017). However, the associated travel distance would havehas to be considerably longer, and the initial clast must be large. For example, abrasion of an initial 25\_-cm sized granitic cobble over a distance of ~8 km
- 505 would result in a size reduction of 10 cm in size and expose a centre with a 10Be concentration that is only 8.5% lower compared to the outer layers (Balco et al., 2008; Sklar et al., 2006). The by-product of abrasion, which typically is of silt or clay size (Sklar et al., 2006), unlikely affects the measured 10Be concentrations, as it is finer than the grain size classes typically analysed (Lukens et al., 2016). We did not observe a control of sediment travel distance in catchments with mixed lithologies. The provenance of distinct grain sizes from different
- 510 lithologies has not resulted in a dominantly positive or negative grain size dependency. Potentiallyssibly, because the positions-spatial arrangement of different lithologies in the landscape areis not necessarily elevation-d-dependent, or because these lithologies produce yield minor differences in grain sizes.

**5.2.36.2.3 Mean Annual Precipitation**

The global compilation suggested an additional control of MAP on grain size-dependent 10Be concentrations. 515 Negative The amount and magnitude of negative grain size trends seems to increase with increasing MAP. The highest percentage of negative grain size dependencies are is predominantly found in arid (<100 mm yr-1) and humid catchments (>2000 mm yr-1).7 whereas Mediterranean and temperate catchments (100 2000 mm yr-1) show no or only a weak grain size dependency. Hhowever this distribution intrend is driven related to -by-a low total number of catchments. Negative MAP may control grain size -dependenciesent 40Be concentrations at 520 higher MAP values in the global compilation through its effect on-could be related to higher denudation rates and the-increasing depth of erosion processes (e.g. precipitation-induced landslides; Chang et al., 2007; Chen et al., 2006; Lin et al., 2008). This conforms todiffers from the results the results from our interpretation of the results case study in from the Chilean Coastal Cordillera, in which also showsrevealed a negative grain size dependency in the most arid catchment (AZ) and no grain size trend in the temperate catchment (NA)which 525 additionally revealed a potential-we emphasize the control of MAP on the thickness of the mixed soil layer resulting in uniform 10Be concentrations in the gently sloping temperate catchment (NA). The climatic gradient in our case study did not include a humid catchment with MAP >2000 mm yr+. The discrepancy with the global compilation may result from the additional effect of hillslope angle, which additionally also affects the influences the thickness of the soil mantle and the depth of erosion processes (Heimsath et al., 2009).-but nNegative grain size dependencies in humid catchments may result from precipitation induced landslides during 530 extreme rain events (Chang et al., 2007; Chen et al., 2006; Lin et al., 2008). We thus think that the observed grain size dependencies in the global compilation could be related to the effect of MAP on the thickness of

**5.36.3 Implications**

- 535 Our results and the above discussion suggest that grain size trends in 10Be concentrations are best explained by the effects of hillslope angle and MAP in the context of on the presence and thickness of mixed soil layers and the scouring depth of hillslope erosion processes.\_\_\_\_\_In large catchments, an additional effect may emerge, resulting from\_by abrasion during transport, which could induce a non-representative grain size distribution.-and sediment provenance processes. At present, however, it is difficult to quantify the relative roles of hillslope
- angle, precipitation, travel distance, and lithology, because these parameters tend to be partly correlated. For example, high and steep topography is often associated with high amounts of orographic precipitation, and long travel distances are associated with high total relief large catchments (Figure S6)., where the chance for multiple lithologies is higher than for smaller catchments.
- In any case, the presumed role that soils and different hillslope erosion processes play for grain size-dependent 10Be concentrations is likely not linearly related to variables like mean hillslope angle or mean annual precipitation. Instead, our results are consistent with the presence of thresholds. Landslides likely become important when hillslope angles exceed a critical threshold value (Burbank et al., 1996) and o. And once precipitation is high enough to sustain vegetation and soils, diffusive processes may dominate gently-sloping and soil-mantled landscapes. Such a threshold control on the occurrence of grain size-dependent 10Be concentrations may be the reason why our multivariate-linear regression statistics, using a linear model, yielded mostly insignificant results or low R2-values (Figure 9 and Table S5). More data may allow better constraining the controls and relative importance of these factors in the future. It additionally highlights the importance of
- We evaluated the likelihood of grain size-dependent 10Be concentrations and a potential bias in previously published 10Be-derived catchment-average denudation rates, by comparing our findings with a recently published global compilation (Codilean et al., 2018). Out of 2537 different catchments with an area <5000 km2, 55.7% have hillslope angles >15°, where our data first shows significant grain size effects, and 23.3% exceeded have the threshold-hillslope angles >25°. When considering sediment travel distances, using the relationship between catchment area and sediment travel distance that emerged from our global compilation (R2= 0.99;

systematic studies on single factors, like our study on the sole effect of MAP in the Chilean Coastal Cordillera.

- 560 Figure S6) about 61.9%, 49.8% and 29.2% of the catchments have exceeded the sediment travel distances of >2300 m, >3500 m and >7000 m, respectively. Finally, 2.8% of the catchments have MAP <100 mm yr-1 and 11.5% of the catchments have MAP >2000 mm yr-1, based on GPCC-derived MAP at the sample location. Therefore, previously published catchment-average denudation rates may more frequently be biased as a result of steep hillslopes and long sediment travel distance and less frequently by the influence of MAP. When
- 565 considering a combined effect of all controlling factors in each catchment (slope >25°, sediment travel distance >7000 m and MAP <100 or >2000 mm yr-1), 489.01% of the catchments are predicted to be devoid of grain size dependencies of 10Be concentrations and biased catchment-average denudation rates, whereas 520.09% might contain a bias because one or more of the controlling factors has exceeded the threshold values that emerged from our study.

**570 6.7. Conclusion**

In this paper, we used a field study in Chile and a global compilation of previously published data to assess in what type of catchments grain size-dependent 10Be concentrations may lead to biased estimates in catchmentaverage denudation rates. Our results suggest that <del>M</del>
[revised manuscript text omitted]
 | Longitude | MAP a | Area  | Mean elevation Total Relief |             | Mean slope b | Mean channel steepness c |
|--------------------|----------------|----------|-----------|------------------|-------|------------------------------------|-------------|-------------------------|-------------------------------------|
|                    |                | (°N)     | (°E)      | (mm yr¹)         | (km²) | (m)                                | (m)  | (°)                     | m 0.9             |
| Pan de Azúcar (AZ) | Arid           | -26.112  | -70.551   | 13               | 0.04  | 339                                | 72   | 8.2                     | 7.1                                 |
| Santa Gracia (SG)  | Semi-arid      | -29.760  | -71.168   | 88               | 0.88  | 773                                | 337  | 17.2                    | 32.2                                |
| La Campana (LC)    | Mediterreanean | -32.954  | -71.069   | 358              | 7.41  | 1323                               | 1535 | 23.1                    | 88.8                                |
| Nahuelbuta (NA)    | Temperate      | -37.808  | -73.014   | 1213             | 5.79  | 1308                               | 306  | 8.9                     | 20.5                                |

a Mean annual precipitation (MAP) is derived estimates derived from the GPCC dataset (Meyer-Christoffer et al., 2015).

b Total mean basin slope calculated with a 30m DEM.

c Normalized channel steepness index.

925

Table 2: Cosmogenic nuclide samples from the Chilean Coastal Cordillera. IGSN number, analyzed quartz mass,  ${}^{9}Be$  carries mass,  ${}^{10}Be/{}^{9}Be$  ratio (±1 $\sigma$ ),  ${}^{10}Be$ 940concentrations (±2 $\sigma$  analytical error), spallation (Psp) and muogenic (Pmu) production rates and calculated denudation rates (±2 $\sigma$ ).

| Catchment          | Grain
size | IGSN             | Quartz
mass | 9 Be
Carrier
mass | 10 Be/ 9 Be
ratio | 10 Be
concentration | P sp                                  | P mu                                  | Denudation
rate |
|--------------------|---------------|------------------|----------------|------------------------------------|--------------------------------------------|-----------------------------------|--------------------------------------------------|--------------------------------------------------|--------------------|
|                    |               |                  |                |                                    | ± 1σ                                       | ± 2σ                              |                                                  |                                                  | ± 2σ               |
|                    | (mm)          |                  | (g)            | (mg)                               | x 10 -14                        | (x 10⁵ atoms
g¹)               | (atoms g qtz -1
yr -1 ) | (atoms g qtz -1
yr -1 ) | (mm kyr¹)          |
|                    | 0.5-1         | GFRD10010 | 9.9            | 0.153                              | 43.0 ± 1.5                                 | 4.48 ± 0.33                       |                                                  |                                                  | $6.04 \pm 0.69$    |
|                    | 1-2           | GFRD10011        | 17.9           | 0.153                              | 79.9 ± 2.8                                 | $4.60 \pm 0.34$                   |                                                  |                                                  | $5.86 \pm 0.67$    |
|                    | 2-4           | GFRD10012        | 18.7           | 0.154                              | 78.6 ± 5.8                                 | 4.36 ± 0.32                       |                                                  |                                                  | 6.21 ± 0.72        |
| Pan de Azúcar (AZ) | 4-8           | GFRD10013        | 18.2           | 0.153                              | 65.3 ± 3.6                                 | $3.69 \pm 0.42$                   | 4.13                                             | 0.085                                            | 7.5 ± 1.1          |
|                    | 8-16          | GFRD10014        | 18.1           | 0.154                              | 55.2 ± 2.1                                 | 3.14 ± 0.24                       |                                                  |                                                  | 8.9 ± 1.0          |
|                    | 16-32         | GFRD10015        | 15.0           | 0.153                              | 40.8 ± 1.5                                 | 2.80 ± 0.21                       |                                                  |                                                  | 10.2 ± 1.1         |
|                    | 32-64         | GFRD10016        | 18.7           | 0.153                              | 57.6 ± 2.0                                 | 3.16 ± 0.22                       |                                                  |                                                  | 8.9 ± 1.0          |
|                    | Mean          | -                | -              |                                    | -                                          | 3.75 ± 0.24                       |                                                  |                                                  | 7.66 ± 0.69        |
|                    | 0.5-1         | GFRD1000Q | 18.7           | 0.154                              | 85.4 ± 2.9                                 | 4.71 ± 0.33                       |                                                  |                                                  | 8.26 ± 0.91        |
|                    | 1-2           | GFRD1000R        | 14.1           | 0.153                              | 55.1 ± 2.3                                 | 4.02 ± 0.34                       |                                                  |                                                  | 9.8 ± 1.2          |
|                    | 2-4           | GFRD1000S        | 13.8           | 0.153                              | 49.0 ± 2.1                                 | 3.62 ± 0.32                       |                                                  |                                                  | 11.0 ± 1.4         |
| Santa Gracia (SG)  | 4-8           | GFRD1000T | 13.8           | 0.153                              | 50.3 ± 2.4                                 | 3.76 ± 0.37                       | 6.02                                             | 0.097                                            | 10.5 ± 1.4         |
|                    | 8-16          | GFRD1000U        | 20.0           | 0.154                              | 82.5 ± 2.7                                 | 4.25 ± 0.29                       |                                                  |                                                  | 9.3 ± 1.0          |
|                    | 16-32         | GFRD1000V        | 19.3           | 0.154                              | 97.0 ± 3.2                                 | 5.17 ± 0.35                       |                                                  |                                                  | 7.48 ± 0.82        |
|                    | 32-64         | GFRD1000W        | 19.5           | 0.154                              | 90.9 ± 3.0                                 | 4.79 ± 0.33                       |                                                  |                                                  | 8.12 ± 0.89        |
|                    | Mean          | -                | -              |                                    | -                                          | 4.33 ± 0.26                       |                                                  |                                                  | 9.21 ± 0.84        |
|                    | 0.5-1         | GFRD1000C        | 19.4           | 0.154                              | 4.98 ± 0.28                                | 0.264 ± 0.030                     |                                                  |                                                  | 257 ± 35           |
|                    | 1-2           | GFRD1000D        | 20.0           | 0.154                              | 3.44 ± 0.20                                | 0.177 ± 0.021                     |                                                  |                                                  | 384 ± 55           |
|                    | 2-4           | GFRD1000E        | 17.0           | 0.154                              | 6.05 ± 0.30                                | 0.366 ± 0.037                     |                                                  |                                                  | 185 ± 24           |
| La Campana (LC)    | 4-8           | GFRD1000F        | 16.9           | 0.154                              | 5.70 ± 0.32                                | 0.348 ± 0.039                     | 9.94                                             | 0.11                                             | 194 ± 27           |
|                    | 8-16          | GFRD1000G | 19.5           | 0.154                              | 12.29 ± 0.54                               | 0.648 ± 0.059                     |                                                  |                                                  | 104 ± 12           |

|                 | 16-32 | GFRD1000H        | 20.0 | 0.154 | 9.69 ± 0.44 | 0.498 ± 0.047     |       |      | 135 ± 17   |
|-----------------|-------|------------------|------|-------|-------------|-------------------|-------|------|------------|
|                 | 32-64 | GFRD1000J | 16.5 | 0.154 | 9.43 ± 0.43 | $0.588 \pm 0.055$ |       |      | 144 ± 14   |
|                 | Mean  | -                | -    |       | -           | 0.413 ± 0.033     |       |      | 200 ± 22   |
| Nahuelbuta (NA) | 0.5-1 | GFRD10002 | 19.8 | 0.154 | 51.4 ± 1.8  | 2.67 ± 0.19       | 10.72 | 0.11 | 26.0 ± 2.8 |
|                 | 1-2   | GFRD10003 | 18.7 | 0.153 | 49.5 ± 2.4  | 2.72 ± 0.27       |       |      | 25.6 ±94.3 |
|                 | 2-4   | GFRD10004 | 18.7 | 0.153 | 51.8 ± 1.9  | 2.84 ± 0.22       |       |      | 24.5 ± 2.8 |
|                 | 4-8   | GFRD10005 | 19.2 | 0.154 | 49.7 ± 1.8  | $2.67 \pm 0.20$   |       |      | 26.1 ± 2.9 |
|                 | 8-16  | GFRD10006 | 20.0 | 0.153 | 56.6 ± 1.9  | 2.90 ± 0.21       |       |      | 23.9 ± 2.6 |
|                 | 16-32 | GFRD10007 | 19.6 | 0.154 | 43.5 ± 1.6  | 2.29 ± 0.18       |       |      | 30.6 ± 3.4 |
|                 | 32-64 | GFRD10008 | 19.6 | 0.153 | 33.5 ± 1.3  | 1.76 ± 0.14       |       |      | 40.2 ± 4.5 |
|                 | Mean  | -                | -    |       | -           | 2.55 ± 0.16       |       |      | 27.4 ± 2.4 |

**14.15.** Figure captions**